# Cilastatin Attenuates Acute Kidney Injury and Reduces Mortality in a Rat Model of Sepsis

**DOI:** 10.3390/ijms26167927

**Published:** 2025-08-16

**Authors:** María Ángeles González-Nicolás, Blanca Humanes, Raquel Herrero, Mario Arenillas, Beatriz López, Antonio Ferruelo, José Ángel Lorente, Alberto Lázaro

**Affiliations:** 1Renal Physiopathology Laboratory, Department of Nephrology, Instituto de Investigación Sanitaria Gregorio Marañón, Hospital General Universitario Gregorio Marañón, 28007 Madrid, Spain; rengac@yahoo.es (M.Á.G.-N.); blanca.humanes@gmail.com (B.H.); 2Department of Physiology, School of Medicine, Complutense University of Madrid, 28040 Madrid, Spain; 3RICORS2040, 28007 Madrid, Spain; 4Department of Critical Care Medicine, Hospital Universitario de Getafe, 28905 Madrid, Spain; rherrero@salud.madrid.org (R.H.); joseangel.lorente@salud.madrid.org (J.Á.L.); 5CIBER de Enfermedades Respiratorias, Instituto de Investigación Carlos III, 28029 Madrid, Spain; antonioferruelo@gmail.com; 6Fundación de Investigación Biomédica del Hospital Universitario de Getafe, 28905 Madrid, Spain; 7Clinical Department, Faculty of Health, Medicine and Sports, Universidad Europea, 28670 Madrid, Spain; 8Department of Bioengineering, School of Biomedical Engineering, Universidad Carlos III, 28903 Madrid, Spain; 9Animal Medicine and Surgery Department, Complutense University of Madrid, 28040 Madrid, Spain; marioare@ucm.es; 10Department of Pathology, Hospital General Universitario Gregorio Marañón, 28007 Madrid, Spain; bealopezmb@gmail.com

**Keywords:** acute kidney injury, sepsis, cilastatin, inflammation, nephroprotection, CLP, TLR4

## Abstract

Sepsis is a life-threatening condition caused by an abnormal host response to infection, leading to organ dysfunction and potentially death. Acute kidney injury (AKI) is a critical complication of sepsis. Various pathways, especially signaling through Toll-like receptors (TLRs) and the nucleotide-binding oligomerization domain, leucine-rich repeat and pyrin domain-containing protein 3 (NLRP3) inflammasome, contribute to inflammation and tissue damage. Cilastatin, a renal dehydropeptidase I inhibitor, has shown promise in protecting against AKI induced by nephrotoxic drugs. This study assessed cilastatin’s effectiveness in preventing AKI and inflammation caused by sepsis and its impact on survival. Sepsis was induced in male *Sprague-Dawley* rats using the cecal ligation puncture (CLP) model, with four groups: sham (control), CLP, sham + cilastatin, and CLP + cilastatin. Cilastatin (150 mg/kg) was administered immediately and 24 h after sepsis induction. Kidney injury was evaluated 48 h later by assessing serum creatinine, blood urea nitrogen, glomerular filtration rate, proteinuria, kidney injury molecule-1 levels, and renal morphology. Inflammatory and fibrotic biomarkers, particularly related to the TLR4 and NLRP3 pathways, were also measured. Cilastatin treatment prevented kidney dysfunction, reduced inflammatory markers, and improved survival by 33%. These results suggest that cilastatin could be a beneficial therapeutic strategy for sepsis-related AKI, improving outcomes and reducing mortality.

## 1. Introduction

Sepsis is a life-threatening organ dysfunction caused by a dysregulated host response to infection that leads to organ failure and, potentially, death. It is a major cause of morbidity and mortality worldwide [1,2]. In the United States, sepsis is the tenth leading cause of death, with approximately 250,000 deaths per year, significantly increasing healthcare costs [3,4].

Acute kidney injury (AKI) is a frequent and major complication of sepsis. It affects approximately half of all patients with sepsis in the intensive care unit, where it is the leading cause of death [4,5,6], with a mortality rate ranging between 50% and 80% [7,8,9]. The pathogenesis of sepsis-induced AKI is multifactorial and involves a combination of pathophysiological mechanisms that impair both renal perfusion—including systemic vasodilation, hypotension, and circulatory collapse—and intrinsic renal function, such as hypoxia, oxidative stress, mitochondrial dysfunction, leukocyte infiltration, and the activation of apoptotic pathways. These processes collectively exacerbate tubular epithelial cell injury, ultimately resulting in a rapid decline in glomerular filtration rate (GFR) [4,5,7,9]. Although the mechanisms and pathways underlying sepsis-induced AKI are not fully understood owing to the complex interconnection of factors during its development, the inflammatory response triggered by sepsis is considered one of the leading causes in the onset and maintenance of AKI [9,10]. Renal infiltration by leukocytes (activated by any pathogen or infection) produces pro-inflammatory cytokines and chemokines, such as tumor necrosis factor-alpha (TNFα), interleukin (IL)-6, and IL-1β, in turn contributing to tubular cell injury, partly through an increase in local oxidative stress, which elicits a rise in reactive oxygen species (ROS) and lowered levels of antioxidants [10]. This spiral of damage is amplified indefinitely by the production of more inflammatory, chemotactic, and transcription factors from damaged renal cells, ultimately resulting in renal dysfunction and death. The mortality of patients with sepsis and AKI is about 75%, while in those with sepsis without AKI mortality ranges between 27% and 32% [11]. Therefore, there is an urgent need to develop innovative and efficacious therapies for treating AKI. While various mechanistic pathways have been implicated in the production of inflammation and tissue injury, including the Fas/Fas ligand system and ROS, signaling via pattern recognition receptors, such as Toll-like receptors (TLRs) and inflammasomes, is currently considered to play a decisive role. Specifically, TLR2 and TLR4 are constitutively expressed in renal cells, and activation of the latter is considered a principal inducer of inflammation and injury in the kidney [12,13]. Therefore, interference with the TLR4 signaling pathway has been proposed as a potential therapeutic target to decrease inflammation and sepsis-induced AKI [13,14,15].

We previously reported in vitro and in vivo data on the beneficial effect of cilastatin against the common nephrotoxic agents used in human medicine, such as cisplatin, cyclosporine, and some antibiotics, with no effect on their therapeutic activity [16,17,18,19,20,21,22,23]. Cilastatin is an inhibitor of dehydropeptidase-I (DHP-I), which is present in the cholesterol rafts of the brush border of renal proximal tubular epithelial cells [17,18,19,20]. Our studies have shown that cilastatin binding to brush border DHP-I disrupts the free movement of cyclized lipid rafts, leading to a reduction in or cancellation of key steps in the extrinsic pathway of apoptosis [17,18,19]. Cilastatin protects against oxidative stress, inflammation, and cell death through apoptosis induced by nephrotoxic agents [24].

In the present study, we evaluate whether treatment with cilastatin exerts protective effects by modulating inflammation and fibrosis against non-toxic kidney injury, such as that induced by sepsis/septic shock. Our findings support the potential usefulness of cilastatin in treating sepsis, as it reduces sepsis-induced AKI by decreasing inflammation and fibrosis and preventing them from becoming fatal.

## 2. Results

### 2.1. Cilastatin Improves Morphology and Renal Function in Rats Undergoing Cecal Ligation Puncture

Forty-eight hours after cecal ligation and puncture (CLP), serum levels of blood urea nitrogen (BUN) and creatinine were significantly elevated, and GFR was reduced compared to the sham-operated control group, reflecting sepsis-induced renal dysfunction (Figure 1A–C). Treatment with cilastatin effectively prevented these alterations. Similarly, proteinuria increased in the CLP group but was significantly reduced in cilastatin-treated animals (Figure 1D).

Histopathological analysis of kidney tissue confirmed these findings. Septic kidneys exhibited pronounced morphological alteration, including extensive formation of hyaline protein casts within renal tubules, brush border loss, tubular dilation, vacuolization, leukocyte infiltration, and widespread glomerular hemorrhage and collapse (Figure 1E). Cilastatin markedly improved renal architecture, preserving tubular integrity and reducing glomerular injury (Figure 1E). Semi-quantitative analysis confirmed the presence of morphological damage in septic kidneys, which was significantly reduced in cilastatin-treated animals (Figure 1F).

Expression levels of kidney injury molecule-1 (KIM-1), a specific marker of kidney damage, obtained by quantifying the labeled area in the images of the samples for each experimental group, were significantly increased in kidneys from septic rats compared to controls (Figure 1G,H). Cilastatin markedly decreased KIM-1 expression in the tubules, indicating milder morphological damage (Figure 1G,H).

Cilastatin did not induce changes in renal function, morphology, or KIM-1 expression in non-septic control animals (Figure 1).

### 2.2. Effect of Cilastatin on Sepsis-Induced Hyperinflammation

#### 2.2.1. Cilastatin Decreases Renal Infiltration of Monocytes/Macrophages and Reduces Adhesion Molecule Expression

A hallmark of sepsis-induced renal inflammation is the infiltration of monocytes, which differentiate into macrophages and drive the production of pro-inflammatory cytokines. To investigate this, immunolocalization of CD68 (a monocyte/macrophage marker) and its chemoattractant protein, monocyte chemoattractant protein-1 (MCP-1), was performed in kidney sections.

As shown in Figure 2 (panels A and B), monocyte/macrophage infiltration was markedly increased in septic animals compared to controls, which corresponded with elevated levels of MCP-1 (Figure 2A,C). Cilastatin treatment significantly reduced both cellular infiltration and MCP-1 expression.

Adhesion molecules, such as vascular cell adhesion molecule-1 (VCAM-1) and intercellular adhesion molecule-1 (ICAM-1), which mediate leukocyte extravasation to inflamed tissues, were also upregulated in the kidneys of septic animals (Figure 2D–F). Cilastatin significantly attenuated this sepsis-induced increase in adhesion molecule expression, changes in line with previous results

Cilastatin alone had not effect on MCP-1 levels, monocyte/macrophage infiltration, or ICAM-1/VCAM-1 expression in non-septic animals compared with the sham-operated control group (Figure 2).

#### 2.2.2. Cilastatin Prevents Activation of the Renal TLR4/Myd88/NF-κB Inflammatory Axis

The TLR4 signaling pathway, activated by bacterial endotoxins, is a major driver of inflammatory injury in sepsis, contributing to inflammasome activation and cell death [15]. Western blotting and immunofluorescence showed markedly elevated cytoplasmic TLR4 expression in kidneys from septic animals compared to controls (Figure 3A,B,D). *TLR4* mRNA expression was also significantly increased following sepsis induction (Figure 3C). Cilastatin treatment significantly reduced both mRNA and protein levels of TLR4 in septic animals (Figure 3A–D).

Lipopolysaccharide (LPS) Binding Protein (LBP), which facilitates LPS binding to TLR4, was significantly elevated in septic animals, in line with increased TLR4 expression and inflammation (Figure 3E). Cilastatin completely suppressed LBP expression (Figure 3E).

Myeloid differentiation factor 88 (MyD88), a central adaptor in the TLR4 pathway and a known contributor to acute kidney injury [25], showed increased protein expression in kidneys from septic rats, as determined through Western blotting (Figure 3F,G) and confirmed through immunohistochemistry/immunofluorescence (Figure 3I,J). Cilastatin normalized MyD88 protein levels (Figure 3A–E). However, no differences in *MyD88* mRNA expression were observed among the groups (Figure 3H).

Nuclear factor kappa B (NF-κB), the downstream effector of TLR4 signaling, regulates numerous inflammatory mediators. As shown in Figure 4, sepsis significantly increased both systemic and renal NF-κB expression, which was attenuated by cilastatin treatment. In addition to the increase in total NF-κB protein levels, elevated phosphorylation of the p65 subunit was observed in the kidneys of the CLP group, as determined through Western blot analysis. This modification enhanced the transcriptional activity of NF-κB, leading to an upregulation of IL-6 levels. Treatment with cilastatin effectively inhibited p65 phosphorylation and, consequently, reduced the serum IL-6 levels regulated by this pathway (Figure 4F).

Cilastatin had no effect on these variables in control animals (Figure 4).

#### 2.2.3. Cilastatin Attenuates NLRP3 Inflammasome Activation in Septic Kidneys

Nucleotide-binding oligomerization domain, leucine-rich repeat and pyrin domain-containing protein 3 (NLRP3) inflammasome is a key regulator of inflammation and pyroptosis, mediating caspase-1 activation and subsequent processing of pro-inflammatory cytokine. In line with findings for TLR4, MyD88, and NF-κB, sepsis significantly increased renal expression of NLRP3 and caspase-1 and its activated form (cleaved caspase-1 protein) compared to controls, as shown through Western blotting (Figure 5A–E).

Cilastatin treatment reduced the expression of these proteins and significantly downregulated *caspase-1* gene expression (Figure 5A–F). Caspase-1 activation leads to the cleavage and release of mature IL-1β. Protein (pro-IL-1β) and mRNA levels of *IL-1β* were significantly elevated in septic kidneys, while cilastatin treatment restored them to levels comparable to sham controls (Figure 5G–I).

Cilastatin did not affect these markers in control animals.

#### 2.2.4. Cilastatin Reduces Sepsis-Induced Expression of Profibrotic Cytokines in the Kidney

Hyperinflammation and impaired tissue repair are associated with renal fibrosis. Connective tissue growth factor (CTGF) and transforming growth factor beta (TGF-β) are key profibrotic cytokines involved in inflammation, tissue remodeling, and fibrogenesis. Immunolocalization revealed significantly increased levels of both CTGF and TGF-β in the kidneys of septic animals compared to sham controls, suggesting early activation of fibrotic pathways (Figure 6A–C).

Although the duration of the model (48 h) was short, collagen fiber staining indicated an early fibrotic response in septic kidneys, albeit without statistically significant differences (Figure 6A,D). Cilastatin treatment significantly reduced CTGF and TGF-β expression, thereby protecting renal tissue from early fibrotic progression (Figure 6).

### 2.3. Cilastatin Increases Survival in Rats with CLP-Induced Sepsis

Mortality is a frequent outcome in sepsis. In this aggressive model, five out of six animals in the untreated CLP group died within 48 h. Cilastatin treatment reduced mortality by 33% (Figure 7).

## 3. Discussion

Sepsis is a life-threatening condition caused by a dysregulated host response to infection and constitutes a mayor global health burden. AKI is a serious complication of sepsis and a leading cause of death among critically ill patients, acting as an independent predictor of increased mortality [9]. Despite numerous experimental therapies demonstrating protective effects against sepsis-induced AKI at the preclinical level [10,26,27], virtually all have failed to translate into clinical application, and no effective treatment is currently available [28]. Thus, there is an urgent need for innovative and effective therapeutic approaches.

In this study, we demonstrate that cilastatin exerts protective effects against AKI caused by polymicrobial infection in a rat model of CLP-induced sepsis. This model closely recapitulates the pathophysiological features and clinical progression of human sepsis and is considered the gold standard for preclinical studies [29,30]. CLP-induced sepsis consistently results in AKI, characterized by elevated serum creatinine and BUN levels and reduced GFR [10,26,27], making it a widely accepted model for investigating novel therapeutic interventions [31].

Cilastatin treatment significantly reduced sepsis-induced elevations in creatinine and BUN and restored renal function [10,26,27]. It also normalized levels of KIM-1, a sensitive early biomarker of AKI that is elevated in sepsis and associated with poor prognosis [10,32]. The functional improvements were corroborated by the preservation of renal histological architecture and a marked reduction in leukocyte infiltration. The renoprotective effects of cilastatin are consistent with its previously reported efficacy in models of nephrotoxic AKI induced by cisplatin, gentamicin [16,17,19,23], rhabdomyolysis, and ischemia-reperfusion injury [33,34].

Multiple overlapping mechanisms contribute to the pathogenesis of sepsis-associated AKI. Among all of them, inflammation and the excessive release of pro-inflammatory cytokine are considered central contributors to renal injury during sepsis [1,2,6,9,10,27,35]. TLRs, particularly TLR4 expressed in proximal tubular epithelial cells, are critical in mediating this response [15,27,36]. Upon activation by bacterial endotoxins, TLR4 initiates an innate immune cascade that promotes immune cell recruitment and activation (including leukocytes, monocytes, and macrophages) and induces the release of pro-inflammatory cytokines (e.g., TNFα, IL-6), chemokines (e.g., MCP-1), and adhesion molecules, culminating in renal injury [15,27,36]. Thus, modulation of TLR4 signaling is considered a promising therapeutic strategy for sepsis-induced AKI [15,36]. Multiple studies support the efficacy of TLR4 inhibition using natural and synthetic compounds [15]. For instance, Zhao H. et al. showed how betulin, a natural triterpenoid, suppressed TLR4 and NF-κB expression and mitigated CLP-induced kidney injury [14]. Similarly, Chlorogenic acid, a constituent of *Lonicerae flos*, reduced levels of TNFα, IL-6, and IL-1β by inhibiting TLR4/NF-κB signaling in LPS-induced AKI [12]. Other compounds, such as SKLB023 and rhein, have also shown nephroprotective effects via anti-inflammatory pathways [27,37].

Our findings reveal that cilastatin exerts similar effects by downregulating TLR4 expression at both mRNA and protein levels and significantly reducing the expression of the adaptor molecule MyD88, which plays a key role downstream of TLR4 and is implicated in AKI pathogenesis [25]. MyD88 has been proposed as a therapeutic target in sepsis-induced AKI [25]. Cilastatin also reduced NF-κB expression at both the renal and the systemic level, confirming previous findings in therapeutic interventions [12,13,14]. NF-κB is a master regulator of inflammation, orchestrating the transcription of numerous inflammatory mediators, including TNFα, interleukins, adhesion molecules, and chemokines, such as MCP-1. It also regulates immune cell activation and their pro-inflammatory secretions [16]. In sepsis, excessive NF-κB activation contributes to a self-perpetuating inflammatory loop, leading to tissue damage, fibrosis, and progressive renal dysfunction.

In parallel to TLR4, the NLRP3 inflammasome—a pattern recognition receptor involved in innate immune responses—also plays a central role in inflammation and tissue injury in sepsis-induced AKI. Overactivation of and disproportionate responses to NLRP3 are involved in the development of numerous conditions, including kidney disease and sepsis-induced AKI [38,39,40,41]. Inhibition of the NLRP3 inflammasome and its products has also been proposed as a therapeutic target in the treatment of acute renal failure, including sepsis-induced AKI [40,41,42].

In our model, CLP-induced sepsis increased the renal expression of NLRP3, caspase-1 in its mature and activated form, and IL-1β, consistent with previous reports. Yanhui Cao et al. demonstrated that genetic deletion of NLRP3 in mice reversed CLP-induced increases in serum creatinine, neutrophil infiltration, and the expression of caspase-1, IL-1β, and IL-18 [43].

Our results support the nephroprotective role of cilastatin via suppression of the NLRP3/caspase-1/IL-1β axis. Notably, the degree of protection observed in our model with cilastatin was greater than that in the genetic knockout model, further supporting the role of cilastatin in upstream inhibition of inflammasome activation through the TLR4/MyD88/NF-κB blockade (i.e., signal 1 priming) [40].

Collectively, the anti-inflammatory effects of cilastatin reduced the hyperinflammatory state associated with sepsis, leading to decreased expression of endothelial adhesion molecules (VCAM-1, ICAM-1), reduced chemokine production, and lowered leukocyte infiltration in the kidney. Similar findings were reported by Wang X et al. using resveratrol, a known antioxidant [44].

Chronic kidney injury and fibrosis are common outcomes of prolonged inflammation. TLR4/NLRP3 signaling has been implicated in fibrogenesis and progression to chronic kidney disease [45,46]. In our model, septic animals exhibited early activation of fibrotic markers TGF-β and CTGF, which were significantly downregulated by cilastatin treatment, thereby halting the progression toward chronic damage.

Previous studies from our group have shown that cilastatin protected the kidney from nephrotoxic agents, such as cisplatin and gentamicin, by attenuating inflammation [16,19]. Specifically, cilastatin reduced NF-κB and TNFα activation, the expression of cytokines and chemokines (IL-6, MCP-1), adhesion molecule production, and monocyte/macrophage recruitment in renal tissue [16,19].

Here, we extend those findings by demonstrating that cilastatin also protects against sepsis-induced AKI via interference with the TLR4/MyD88/NF-κB and NLRP3 signaling pathways, resulting in improved survival. However, the observed survival benefit may not be solely attributed to renal protection. Choudhury et al. showed that inhibition or deletion of DHP-1 reduced neutrophil infiltration in the lungs and liver and improved survival in endotoxemic models [47]. Similarly, our previous work has demonstrated cilastatin’s protective effects in LPS-induced lung injury [48]. Thus, multiorgan protection may underlie the improved survival rates, although renal preservation is likely a key contributor.

Cilastatin was originally developed to inhibit the renal metabolism of imipenem by DHP-I, and it is not a known anti-inflammatory agent or direct ligand of TLR4 or NLRP3, although it protects against sepsis-induced AKI by decreasing inflammation.

Our prior findings showed that cilastatin binds to DHP-I located in cholesterol-rich lipid rafts, altering membrane fluidity and potentially disrupting raft-associated signaling processes that protect tubular cells [16,17,18,19,20,21]. This was evidenced by reduced internalization of raft-associated proteins, such as cholera B-toxin receptor, the Fas/FasL death complex, and gentamicin via megalin [17,19,21].

The integrity of cholesterol-rich rafts is essential for TLR4 signaling [49]. TLR4 and its co-receptors (CD14, CD44, CD36) localize to these domains, and disruption of the raft structure impairs pathway activation [49]. Cholesterol-depleting agents like cyclodextrin and nystatin have been shown to inhibit assembly of the TLR4 complex and block LPS-induced TNFα production [49,50]. Cilastatin’s binding to DHP-I within the same raft microdomains likely alters raft fluidity, inducing conformational and steric changes that interfere with TLR4 complex formation and activation (Figure 8).

This mechanism may also explain the observed reduction in serum LBP levels following cilastatin treatment. While LBP typically facilitates the transfer of LPS to CD14, the subsequent transfer to TLR4 appears to be impaired due to alterations in cholesterol raft integrity. This disruption likely inhibits the assembly and activation of the TLR4 signaling complex and its downstream components. Similar findings were reported by Szabo et al., who demonstrated that ethanol interferes with LPS-induced redistribution of TLR4 complex components within lipid rafts, preventing CD14-mediated activation and, consequently, attenuating inflammation and tissue injury [51].

Our study has several limitations, primarily due to the inherent difficulty of replicating the clinical syndrome of sepsis under experimental conditions. Although we employed the CLP model, which is widely regarded as the gold standard for sepsis research, it is not standardized across laboratories. Variations in surgical technique and postoperative care contribute to inconsistencies when comparing results between studies. Additionally, differences in the composition of the intestinal microbiota among individual animals or species may also account for variability in outcomes. In our study, we were unable to identify the specific microbial species involved in the development of sepsis in the polymicrobial CLP model, nor could we determine whether cilastatin exerted any effects at this level.

In conclusion, our findings support the potential use of cilastatin as a protective agent in sepsis-induced AKI. Cilastatin effectively reduced hyperinflammation and early fibrotic signaling, leading to improved renal function and enhanced survival. These nephroprotective effects were mechanistically linked to the downregulation of the TLR4/MyD88/NF-κB and the NLRP3 inflammatory pathways, resulting in attenuation of the systemic and renal inflammatory response and the prevention of fibrosis. Taken together, these results highlight cilastatin as a promising therapeutic candidate for clinical use in patients with sepsis at risk of kidney injury.

## 4. Materials and Methods

### 4.1. Drugs

Crystalline cilastatin (sodium salt of [R-[R, S-(Z)]]-7-[(2-amino-2-carboxyethyl)thio]-2-[[(2,2-dimethylcyclopropyl)carbonyl]amino]-2-heptenoic acid) with molecular formula C_16_H_25_N_2_NaO_5_S (Figure 9) was provided by ACS Dobfar (Milan, Italy) and dissolved in 0.9% saline (vehicle).

### 4.2. Animals

Studies were performed on 8–9-week-old male *Sprague-Dawley* rats weighing 320–370 g (Charles River Laboratories, Wilmington, MA, USA). The animals were housed under controlled light (12 h light–dark cycle), temperature, and humidity with free access to food and water and pre-conditioned in metabolic cages on alternate days for 1 week. The study was approved by the Ethics Committee for Animal Experimentation of Getafe University Hospital, Madrid, Spain (protocol code 003/2015, date of approval: 2015). The animals were handled at all times according to the applicable laws in Spanish Royal Decree 53/2013 and European Directive 2010/63/UE on the protection of animals used for experimentation and other scientific purposes, including teaching, under the direct supervision of the veterinary surgeon in charge.

### 4.3. Induction of Sepsis: Cecal Ligation and Puncture Model

Sepsis was induced following the CLP, as previously described [29,52]. Briefly, animals were anesthetized with intraperitoneal (ip) ketamine (Ketolar 50 mg/mL, ref. 631028, PARKE-DAVIS S.L., Madrid, Spain) 90 mg/kg and ip diazepam (10 mg/mL, ref. 582606, Ecuphar, Barcelona, Spain) 5 mg/kg. A 3 cm midline laparotomy was performed under sterile conditions to expose the cecum, and a 4–0 silk ligature was placed 0.75 cm from the cecal tip. The tip was punctured once with a 14-gauge needle. A second ligature was placed 1.5 cm proximal to the first one and distal to the ileocecal valve to avoid intestinal obstruction. Three punctures were made in the cecum between the 2 ligatures with a 16-gauge needle. The cecum was squeezed gently until a 1 mm column of fecal material was exteriorized. The bowel was then returned to the abdomen and the incision was closed in layers. A sham operation (laparotomy and cecal exposure with no other manipulation) was performed as a control. At the end of surgery, all rats were resuscitated with normal 0.9% saline serum (50 mL/kg) administered subcutaneously (sc) in a single dose, and they received analgesic treatment with buprenorphine (Buprenodale 0.03 mg/mL, ref. 585314, Dechra, Bladel, Netherlands) 0.05 mg/kg sc every 12 h immediately after surgery and until euthanasia.

### 4.4. Experimental Design

The animals were randomly allocated to 4 groups (*n* = 10 in each group): sham-operated (control), CLP, sham-operated + cilastatin, and CLP + cilastatin. Rats were treated with cilastatin 150 mg/kg ip immediately and at 24 h after CLP or sham surgery and placed in metabolic cages for 24 h urine collection. Cilastatin was substituted for vehicle (saline, 0.5 mL/100 g) in the other groups. The dose of cilastatin was selected based on previous experiments where it was shown to be effective in the treatment of acute kidney failure induced by nephrotoxic agents [6,17,19,23].

Forty-eight hours after CLP, all rats were anesthetized with ketamine (10 mg/kg) and diazepam (4 mg/kg) ip and euthanized. Blood samples were collected through the insertion of a cannula into the abdominal aorta, and serum was separated and stored for biochemistry. The kidneys were perfused with cold saline and quickly removed. Kidney samples were snap-frozen in liquid nitrogen and kept at −80 °C or fixed in 4% paraformaldehyde (ref. sc-281692, Santa Cruz Biotechnology, Houston, TX, USA) (24 h) and paraffin-embedded for subsequent analysis.

Another experimental model was performed to determine the mortality/survival of the septic process at 48 h after treatment with or without cilastatin. Sepsis was induced based on the procedure described above, with the difference that the 3 punctures in the cecum between the 2 ligatures were made with a 14-gauge needle and the other puncture (in the cecal tip) was made with a 16-gauge needle. This led to a more aggressive model of sepsis. A total of 12 animals were used. These were randomized into 2 groups with a sample size of 6 animals per group.

The study groups and their treatments were therefore as follows:-CLP (sepsis): Animals with induction of aggressive sepsis through CLP surgery plus saline (in the same manner and volume as in the previous group CLP).-CLP + cilastatin: Animals with induction of aggressive sepsis through CLP plus cilastatin 150 mg/kg immediately and at 24 h after CLP surgery.

At 48 h after CLP, the surviving animals in both groups were counted.

### 4.5. Kidney Function Monitoring

BUN and serum and urine creatinine levels were automatically measured using a Cobas^®^ 6000 analyzer (F. Hoffmann-La Roche Ltd., Basel, Switzerland) following the manufacturer’s protocols. GFR was calculated according to creatinine clearance, and total protein content in urine was quantified using the sulfosalicylic acid method [17].

### 4.6. Renal Histopathology Analysis

For light microscopy, paraffin-embedded renal sections (4 µm thick) were stained with hematoxylin–eosin (ref. 51275, Sigma-Aldrich, St Louis, MO, USA) and Direct Red 80 (Sirius Red, (ref. 365548, Sigma Aldrich). The kidney injury score and the percentage of collagen fibers stained with Sirius Red were evaluated as an index of fibrosis by a pathologist in a blinded fashion, as reported previously [19].

### 4.7. Western Blot and Immunohistochemistry Techniques

Both techniques were performed as previously described [16,17,19]. The primary antibodies used in immunohistochemistry were as follows: goat anti-KIM-1 polyclonal antibody (R&D Systems, Minneapolis, MN, USA [dilution 1:20], ref. AF3689); goat anti-MCP-1 (R17) polyclonal antibody (Santa Cruz Biotechnology, Santa Cruz, CA, USA [dilution 1:50], ref. sc-1785); mouse anti-CD68 monoclonal antibody (AbD Serotec, Oxford, UK [dilution 1:25] ref. MCA341R); goat anti-VCAM-1 (C-19) polyclonal antibody (Santa Cruz Biotechnology [dilution 1:100], ref. sc-1504); mouse anti-ICAM-1 (G-5) monoclonal antibody (Santa Cruz Biotechnology [dilution 1:50], ref. sc-8439); rabbit anti-TGF-1β (V) polyclonal antibody (Santa Cruz Biotechnology [dilution 1:250], ref. sc-146); goat anti-CTGF (L-20) polyclonal antibody (Santa Cruz Biotechnology [dilution 1:100], ref. sc-14939); and rabbit anti-MyD88 polyclonal antibody (Novus Biologicals, Minneapolis, MN, USA [dilution 1:250], ref. NB100-56698). The specificity of the different antibodies was verified by controls lacking the primary antibody, thus producing no background. Evaluation of the surface of the labeled area of all antibodies was assessed using quantitative image analysis, as previously reported [16,17,19].

The primary antibodies used for the Western blot analysis were as follows: mouse anti-RelA/NF-κB p65 monoclonal antibody (Santa Cruz Biotechnology [dilution 1:200], ref. sc-8008); rabbit anti-Phospho-NF-κB p65 (Ser536) (93H1) monoclonal antibody (Cell Signaling Technology, Danvers, MA, USA [dilution 1:1000], ref. 3033); mouse anti-TLR4 (25) monoclonal antibody (Santa Cruz Biotechnology [dilution 1:200], ref. sc-293072); rabbit anti-MyD88 polyclonal antibody (Novus Biologicals [dilution 1:500], ref. NB100-56698); rabbit anti-NLRP3 inflammasome monoclonal antibody (Abcam, Cambridge, UK [dilution 1:300], (ref. ab214185); rabbit anti-IL-1β polyclonal antibody (Abcam [dilution 1:300], ref. ab9722); mouse anti-caspase-1 (14F468) monoclonal antibody (Santa Cruz Biotechnology [1:1000], ref. sc-56036); and rabbit anti-caspase 1 polyclonal antibody (Abcam [dilution 1:3000], ref. ab108362). As an internal standard, membranes were reprobed with a goat anti-voltage-dependent anion channel (VDAC)-1 (N-18) polyclonal antibody (Santa Cruz Biotechnology [dilution 1:2000], ref. sc-8828) to verify the equal loading of protein in each line. All signals were visualized using an Alliance 4.7 instrument (Uvitec, Cambridge, UK) and analyzed using densitometric scanning with ImageJ (Image Processing and Analysis in Java) software, https://imagej.net/ij/index.html, accessed on 1 July 2024 [16,17,19]. The results were expressed as arbitrary units (a.u.).

### 4.8. Immunofluorescence

Paraffin-embedded kidney tissue sections (4 μm) were deparaffinized and rehydrated in decreasing concentrations of alcohol (ref. 20821.365, VWR, Philadelphia, PA, USA). Samples were then heated in 10mM sodium citrate buffer, pH 6 (to boiling for unmasking of the antigens), washed in phosphate-buffered saline (ref. sc-362182, Santa Cruz Technology, Santa Cruz, CA, USA) and Tween 20 (ref. P9416, Sigma Aldrich) 0.1% (PBS-T), and blocked at room temperature (RT) for 1 h in bovine serum albumin (ref. A2153, Sigma Aldrich, St Louis, MO, USA) 4% and host serum (6–10%), in which the secondary antibody was obtained in PBS-T. Sections were then incubated overnight at 4 °C in a humidified chamber with primary antibodies diluted in PBS-T, 4% bovine serum albumin, and 1% host serum in which the secondary antibody was obtained as follows: mouse anti-RelA/NF-κB p65 monoclonal antibody (Novus Biologicals, Minneapolis, MN, USA [dilution 1:100], ref. NB100-56712); mouse anti-TLR4 (25) monoclonal antibody (Santa Cruz Biotechnology [dilution 1:50], ref. sc-293072); and rabbit anti-MyD88 polyclonal antibody (Novus Biologicals [dilution 1:50], ref. NB100-56698). After washing with PBS-T, tissue sections were incubated with DyLight 488-conjugated donkey anti-mouse IgG (H+L) (Novus Biologicals [dilution 1:4000], ref. NBP1-72931), FITC-conjugated goat anti-rabbit (Bethyl Laboratories, Houston, TX, USA [dilution 1:50], ref. A120-201F), or DyLight^®^ 594-conjugated donkey anti-rabbit IgG (Bethyl Laboratories, Houston, TX, USA [dilution 1:40], ref. A120-208D4) secondary antibodies for 1 h in the dark. After washing with PBS, the samples were incubated with a 0.3% solution of Sudan Black B (ref. 199664, Sigma-Aldrich) dissolved in isopropanol (ref. 470610010, Thermo Fisher Scientific, Waltham, MA, USA) for 20 min at RT. After washing, cell nuclei were counterstained with 4′,6-diamidino-2-phenylindole dihydrochloride (DAPI, 5 µg/mL, ref. D8417, Sigma Aldrich). Finally, the slices were mounted with fluorescence mounting medium (ref. S302380-2, Dako, Carpinteria, CA, USA). Immunolocalizations were examined with a Leica-SP2 confocal microscope (Leica Microsystems, Wetzlar, Germany).

### 4.9. Inflammatory Molecules mRNA Expression

*TLR4*, *Myd88*, *caspase-1*, and *IL-1β* gene expression were analyzed using real-time polymerase chain reaction (PCR), as previously described [16,19]. Total RNA was isolated from kidney homogenate using TRIzol Reagent (ref. 12034977.2, Life Technologies, Invitrogen, Paisley, UK), according to the manufacturer’s instructions. In brief, 2 µg total RNA was reverse-transcribed using the iScript cDNA Synthesis Kit (ref. 1708891, Bio-Rad, Hercules, CA, USA). Target gene expression was quantified using Kapa Syber Fast qPCR Kit Master Mix (ref. KK4607, Kapa Biosystems, Wilmington, MA, USA) in the iQ5 Multicolor Real Time PCR Detection System (Bio-Rad). The primer sets used were as follows: *TLR4*, forward GAGGACTGGGTGAGAAACGA and reverse CACCAACGGCTCTGGATAAA; *Myd88*, forward GCCTTGTTAGACCGTGAGGA and reverse CCCAGTTCCTTTGTCTGTGG; *IL-1β*, forward CACCTCTCAAGCAGAGCACAG and reverse GGGTTCCATGGTGAAGTCAAC; and *caspase-1*, forward CCGTGGAGAGAAACAAGGAG and reverse GGTGTTGAAGAGCAGAAAGCA. *Glyceraldehyde 3-phosphate dehydrogenase* (*GAPDH*, forward CGGCCGAGGGCCCACTAAAG and reverse TGCTCAGTGTTGGGGGCTGAGT) served as a housekeeping gene and was amplified in parallel with the genes of interest. The expression of each target gene was normalized to the *GAPDH* signal. All primers were obtained through a literature review and using PRIME3R software (https://primer3.ut.ee/ (accessed on 26 October 2023)) and supplied by Life Technologies, Invitrogen. The Δ threshold cycle (ΔCT) approach was applied for the quantitative cDNA analysis [16,19]. All of the measurements were carried out in duplicate. The results for the controls carried out with distilled water were negative in all tests.

### 4.10. Measurement of Serum LBP, IL6, and NF-κB

LBP, NF-κB, and IL-6 were measured in serum samples using the Rat Lipopolysaccharide Binding Protein (LBP) ELISA Kit (ref. CSB-E11184r), Rat Nuclear factor-kappa B (NF-κB) ELISA Kit (ref. CSB-E13148r), and Rat Interleukin 6, IL-6 ELISA KIT (ref. CSB-E04640r), respectively, all purchased from Cusabio Technology (Houston, TX, USA) and used according to the manufacturer’s indications. The concentration of each sample was quantified through interpolation of the absorbance reading on a standard curve generated with the standards provided by the kits and expressed in pg/mL

### 4.11. Data Analysis

The data analysis was performed using SPSS 11.5 (SPSS, Chicago, IL, USA), and the graphs of the figures were drawn with GraphPad Prism 9.0 (GraphPad Software, Boston, MA, USA). Quantitative variables are presented as mean ± standard error of the mean (SEM). Equality of variances was tested with Levene’s text. Normally distributed continuous variables with equal variances were analyzed using analysis of variance (ANOVA) with the Fisher’s least significant difference (LSD) test as a post hoc analysis to determine specific group differences. When variances were not equal, the Kruskal–Wallis test was applied. Differences were considered statistically significant for bilateral *α*-values < 0.05.

## 5. Patents

M.A.G-N.G., B.H., and A.L. are the coinventors of cilastatin patents for the treatment of sepsis (“Cilastatin for use in the treatment of sepsis”. Number of patents granted: EP 3474897A1; US 11185522; 7109791; CN109843330B; 20177281744. Application for grant in Canada nº: 3,028,846). These were assigned to the Fundación para la Investigación Biomédica del Hospital Gregorio Marañón (FIBHGM), licensed to Telara Pharma S.L., and transferred to Arch Biopartners.

## Figures and Tables

**Figure 1 ijms-26-07927-f001:**
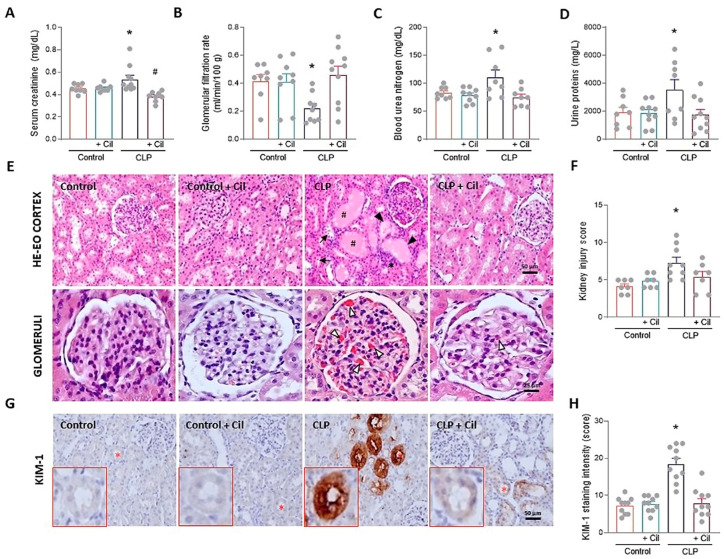
Effects of cilastatin on cecal ligation and puncture (CLP)-induced nephrotoxicity in rats. Parameters of renal function and histology after 48 h of CLP surgery. Animals were treated with 150 mg/kg cilastatin (or its vehicle) once a day for two days after CLP. (**A**) Serum creatinine, (**B**) glomerular filtration rate (GFR), (**C**) blood urea nitrogen (BUN), and (**D**) urine proteins. Results are expressed as mean ± SEM, *n* = 8–10 animals per group, * *p* ≤ 0.0234 vs. all groups; # *p* < 0.04 vs. Control + Cil. (**E**) Representative images of the renal pathology (hematoxylin–eosin staining): upper images (magnification 20×, bar 50 µm); lower images, detail of glomeruli (magnification 40×, bar 25 µm). Signs of sepsis-induced acute kidney injury are marked, including #, accumulation of protein casts in the renal tubules; *, leukocyte infiltration; ▼, loss of brush border membrane; →, blebbing and cell debris detachment; Δ, capillary congestion/glomerular hemorrhage. (**F**) Semi-quantitative kidney injury score, analyzing parameters of glomerular, vascular, and tubule-interstitial damage. Data represent means ± SEM, *n* = 7–9 animals per group, * *p* ≤ 0.034 vs. all groups. (**G**) Immunolocalization of kidney injury molecule-1 (KIM-1) in kidney sections (magnification 20×, bar 50 µm). Red asterisk represents the enlarged tubule in the lower left area of the images. (**H**) Semi-quantitative KIM-1 score. For each sample, the mean staining area was obtained through analysis of the entire sample using Scion image software, version 4.0 (downloaded from https://scion-image.software.informer.com/4.0/ accessed on 1 July 2024). Data represent means ± SEM, *n* = 10 animals per group, * *p* ≤ 0.0001 vs. all groups. All data were analyzed using ANOVA. Cil: cilastatin.

**Figure 2 ijms-26-07927-f002:**
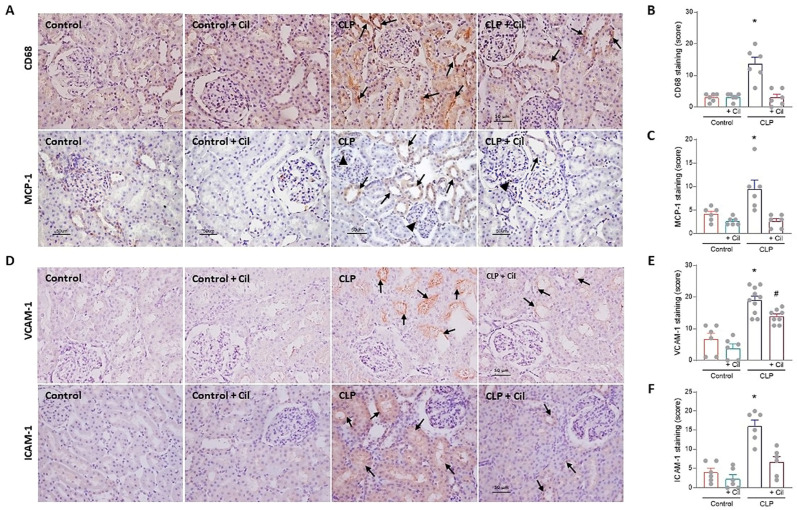
Cilastatin reduces cecal ligation and puncture (CLP)-induced inflammation and expression of adhesion molecules in renal tissue. (**A**) Immunolocalization of CD68 (monocyte/macrophage) (upper image) and monocyte chemoattractant protein-1 (MCP-1) (lower image) in kidney sections. Note the increased staining for both proteins in CLP rats at the tubular level (arrows) and in the glomerular zone (arrowheads) compared with CLP + cilastatin and control rats (magnification 20×, bar 50 µm). (**B**) Immunostaining semi-quantification of CD68 and (**C**) MCP-1 in renal tissue. Results are expressed as mean ± SEM, *n* = 6 animals per group, * *p* ≤ 0.021 vs. all groups. (**D**) Immunolocalization of vascular cell adhesion molecule-1 (VCAM-1) (upper image) and intercellular adhesion molecule-1 (ICAM-1) (lower image) in kidney sections. Note the increased staining for both proteins in CLP rats (arrows) compared with CLP + cilastatin and control groups (magnification 20×, bar 50 µm). (**E**) Semi-quantification of VCAM-1 and (**F**) ICAM-1 immunostaining in renal cells. Results are expressed as mean ± SEM, *n* = 6–10 animals per group, * *p* ≤ 0.0071 vs. all groups; # *p* < 0.001 vs. Control ± Cil. Cilastatin significantly reduced the increase in CD68, MCP-1, VCAM-1, and ICAM-1 previously increased by CLP. The data were analyzed using ANOVA. Cil: cilastatin.

**Figure 3 ijms-26-07927-f003:**
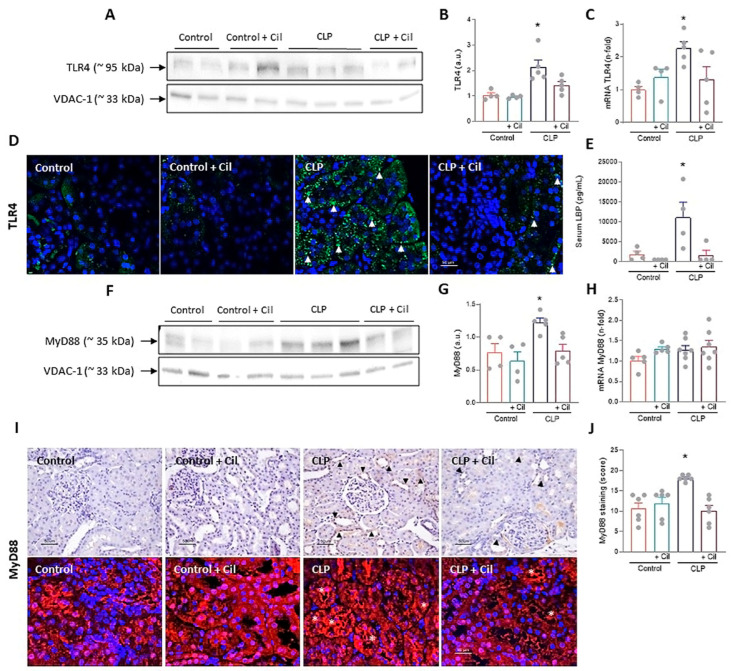
Effects of cilastatin on cecal ligation and puncture (CLP)-induced Toll-like receptor 4 (TLR4)/Myeloid differentiation factor 88 (MyD88) pathway activation. (**A**) Western blot of TLR4 in the renal cortex and (**B**) respective quantification corrected by the loading control. Results are expressed as mean ± SEM, *n* = 4–5 animals per group, * *p* ≤ 0.0094 vs. all groups. (**C**) Renal mRNA expression of *TLR4*. Results are expressed as mean ± SEM, *n* = 4–5 animals per group, * *p* ≤ 0.046 vs. all groups. (**D**) Immunolocalization of TLR4 through immunofluorescence staining (green) in kidney sections (magnification 20× bar 50 µm). Cilastatin significantly decreased the TLR4 levels that were increased by CLP (arrowheads). (**E**) Serum measurement of LPS Binding Protein (LBP). Results are expressed as mean ± SEM, *n* = 4 animals per group, * *p* ≤ 0.0074 vs. all groups. (**F**) Western blot of MyD88 in renal cortex and (**G**) densitometric analysis corrected by the loading control. Results are expressed as mean ± SEM, *n* = 4–5 animals per group, * *p* ≤ 0.0094 vs. all groups. (**H**) Renal mRNA expression of *MyD88*. Results are expressed as mean ± SEM, *n* = 5–7 animals per group. (**I**) Immunolocalization (upper image) and immunofluorescence staining (lower image) of MyD88 in renal tissue (magnification 20× bar 50 µm) and (**J**) semi-quantification. Results are expressed as mean ± SEM, *n* = 6 animals per group, * *p* ≤ 0.0017 vs. all groups. Note increased the staining in CLP rats (arrowhead and asterisks) and the change in the pattern of that staining compared with CLP + cilastatin and control groups. Cilastatin significantly decreased the MyD88 levels that were increased by CLP, although no significant differences in RNA levels were found. The data were analyzed using ANOVA. Cil: cilastatin; a.u., arbitrary units.

**Figure 4 ijms-26-07927-f004:**
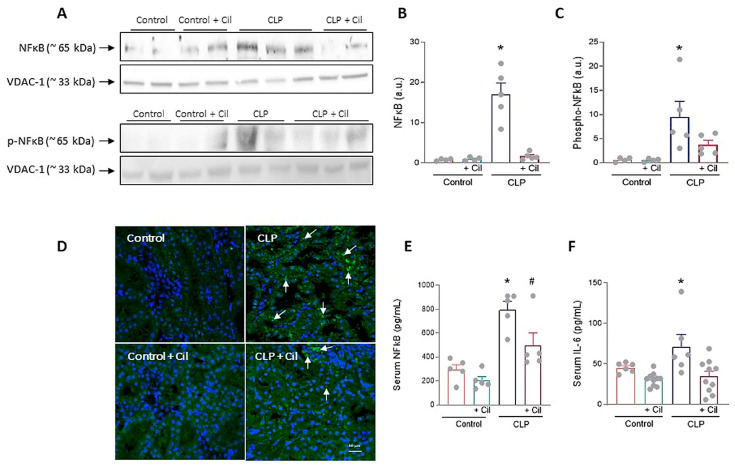
Effects of cilastatin on cecal ligation and puncture (CLP)-induced renal nuclear factor-ĸB (NF-κB) activation and interleukin (IL)-6. (**A**) Representative Western blot of NF-ĸB and Phospho-NF-κB p65 in the renal cortex. (**B**,**C**) Respective quantification corrected by the loading control shows increased NF-κB (* *p* ≤ 0.0001 vs. all groups) and Phospho-NF-κB p65 (* *p* ≤ 0.0412 vs. all groups) in CLP rats compared with CLP + cilastatin and control groups. Data represent means ± SEM, *n* = 4–5 animals per group. (**D**) Localization of NF-ĸB through immunofluorescence staining (green) in kidney sections (magnification 20×, bar 50 µm). An increase in staining can be observed in the CLP group (arrows). (**E**) Serum measurement of NF-ĸB. Data represent means ± SEM, *n* = 5 animals per group, * *p* ≤ 0.0067 vs. all groups; # *p* ≤ 0.022 vs. Control + Cil. (**F**) Serum measurement of IL-6. Data represent means ± SEM, *n* = 6–10 animals per group, * *p* ≤ 0.033 vs. all groups. CLP enhances both the expression and activation of NF-ĸB in renal tissue and raises the serum levels of NF-ĸB and IL-6. Cilastatin significantly reduced the increase in both molecules. The data were analyzed using ANOVA. Cil: cilastatin; a.u., arbitrary units.

**Figure 5 ijms-26-07927-f005:**
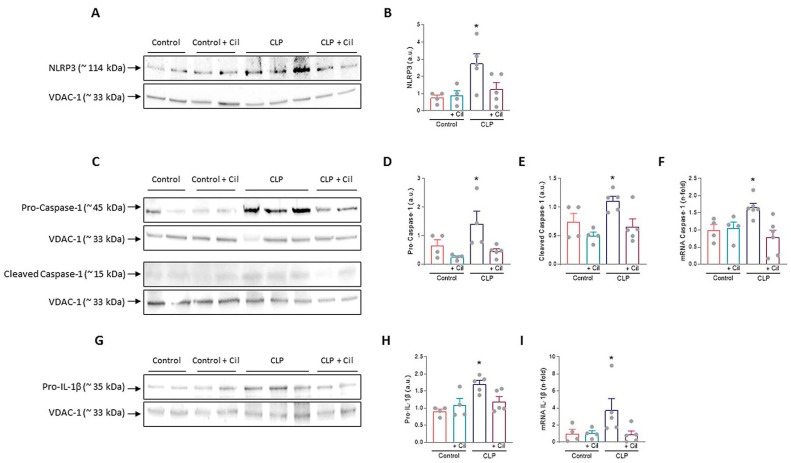
Cecal ligation and puncture (CLP)-induced nucleotide-binding oligomerization domain, leucine-rich repeat and pyrin domain-containing protein 3 (NLRP3) inflammasome activation. Representative images of Western blot in the renal cortex of (**A**) NLRP3, (**C**) Pro- and cleaved caspase-1, and (**G**) Pro-interleukin (IL)-1β. (**B**,**D**,**E**,**H**) Respective Western blot quantifications corrected by loading control shows increased NLRP3 (* *p* ≤ 0.0179 vs. all groups), pro-caspase-1 (* *p* ≤ 0.0486 vs. all groups), cleaved caspase-1 (* *p* ≤ 0.0376 vs. all groups), and pro-IL-1β (* *p* ≤ 0.0134 vs. all groups) in CLP rats compared with CLP + cilastatin and control groups. Renal mRNA expression of (**F**) *Caspase-1* and (**I**) *IL-1β*. Cilastatin significantly decreased the caspase-1 (* *p* ≤ 0.0263 vs. all groups) and IL-1β (* *p* ≤ 0.0388 vs. all groups) mRNA levels increased by CLP in all cases. All results are expressed as mean ± SEM; *n* = 4–5 animals per group. All data were analyzed using ANOVA. Cil: cilastatin; a.u., arbitrary units.

**Figure 6 ijms-26-07927-f006:**
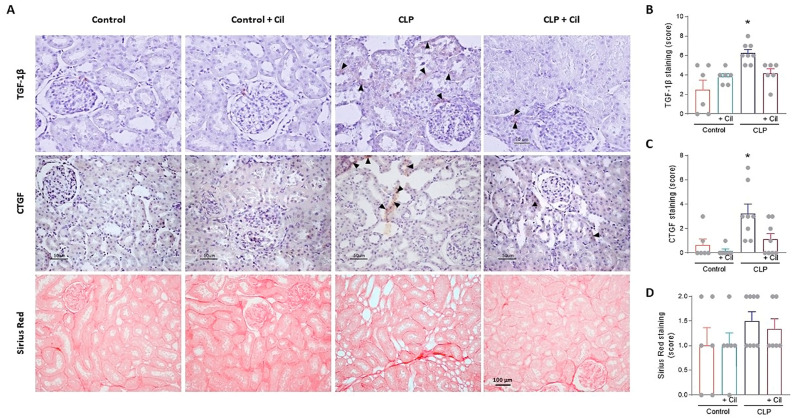
Cilastatin reduces cecal ligation and puncture (CLP)-induced fibrosis. (**A**) Immunolocalization of transforming growth factor beta (TGF-1β) (upper images), connective tissue growth factor (CTGF) (middle images), and Sirius Red staining (lower images) in kidney sections. Note the increased tubular staining in CLP rats (arrowheads) in both factors TGF-1β and CTGF compared with CLP + Cilastatin and control rats (magnification 20×, bar 50 µm for TGF-1β and CTGF, and 100 µm for Sirius Red). Semi-quantification in renal tissue of (**B**) TGF-1β (* *p* ≤ 0.018 vs. all groups), (**C**) CTGF (* *p* ≤ 0.0095 vs. all groups), and (**D**) Sirius Red (the values are not statistically significant), respectively. All results are expressed as mean ± SEM; *n* = 6–8 animals per group. All data were analyzed using ANOVA. Cil: cilastatin.

**Figure 7 ijms-26-07927-f007:**
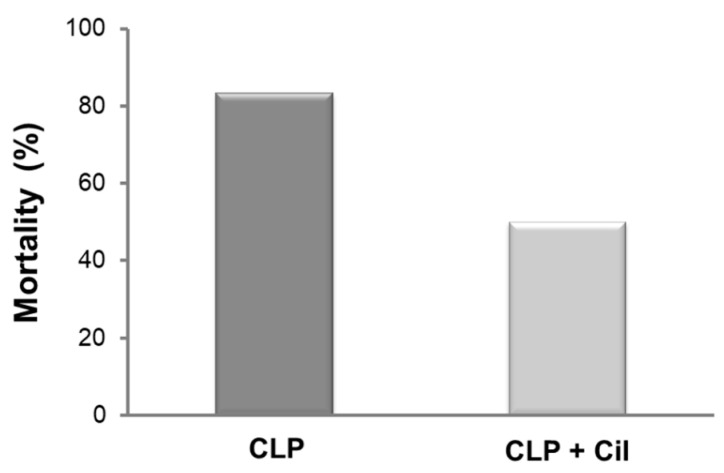
Rat mortality rates after cecal ligation and puncture (CLP). Cilastatin treatment improves survival in rats subjected to CLP-induced sepsis. Mortality was monitored for 48 h after the CLP procedure. There were 6 rats per group. Cil: cilastatin.

**Figure 8 ijms-26-07927-f008:**
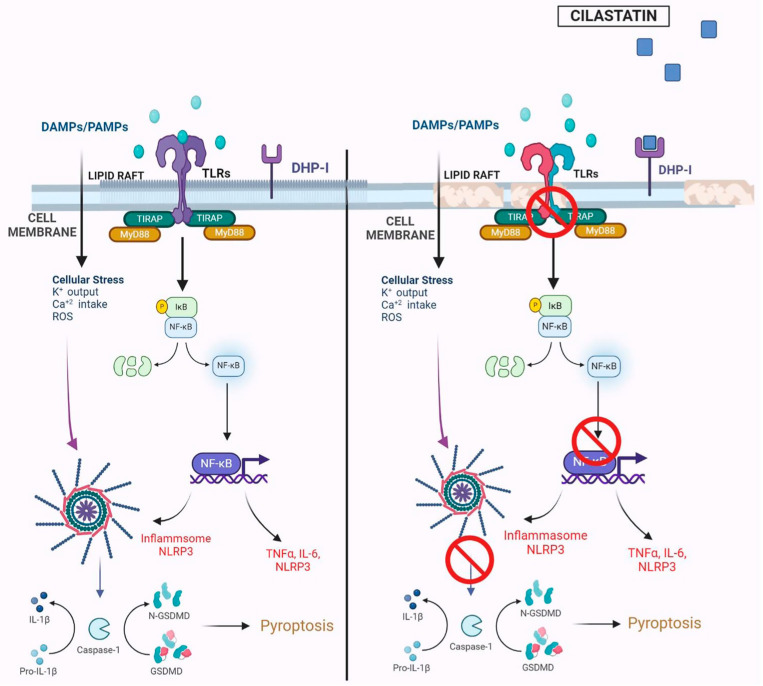
Summary image of the effects of cilastatin on the Toll-like receptor 4 (TLR4)/Myeloid differentiation factor 88 (MyD88)/Nuclear Factor-κB/NF-κB) pathway and nucleotide-binding oligomerization domain, leucine-rich repeat and pyrin domain-containing protein 3 (NLRP3) inflammasome in cecal ligation and puncture (CLP)-induced inflammation and acute kidney injury. The TLR4 signaling complex is organized and localized in the cholesterol lipid rafts on the brush border apical side of renal proximal tubular epithelial cells, geographically close to the renal dehydrodipeptidase I (DHP-I) enzyme. In the left panel, with cholesterol rafts intact, damage-associated molecular patterns (DAMPS) or pathogen-associated molecular patterns (PAMPS), such as bacterial lipopolysaccharide (LPS), activate the TLR4/MyD 88 pathway, leading to the activation of NF-κB and pro-inflammatory cytokine and chemokine production. NF-κB is also involved in the activation of the NLRP3 inflammasome that leads to the activation of caspase-1, which, in turn, regulates the processing of inflammatory pro-cytokines to active cytokines (such as interleukin (IL)-1β), amplifying inflammatory damage and cell death, which exacerbates kidney injury. In the right panel, cilastatin binding to the membrane DHP-I in cholesterol lipid rafts causes modifications in the rafts, preventing the correct assembly and activation of the TLR4 complex and reducing the activation of NF-κB and the NLRP3 inflammasome. Thus, the renal cell is protected. K+, potassium; Ca2+, calcium; TIRAP, TIR Domain Containing Adaptor, Protein; IκB, inhibitor of nuclear factor kappa B; TNFα, tumor necrosis factor alpha; GSDMD, Gasdermin D.

**Figure 9 ijms-26-07927-f009:**
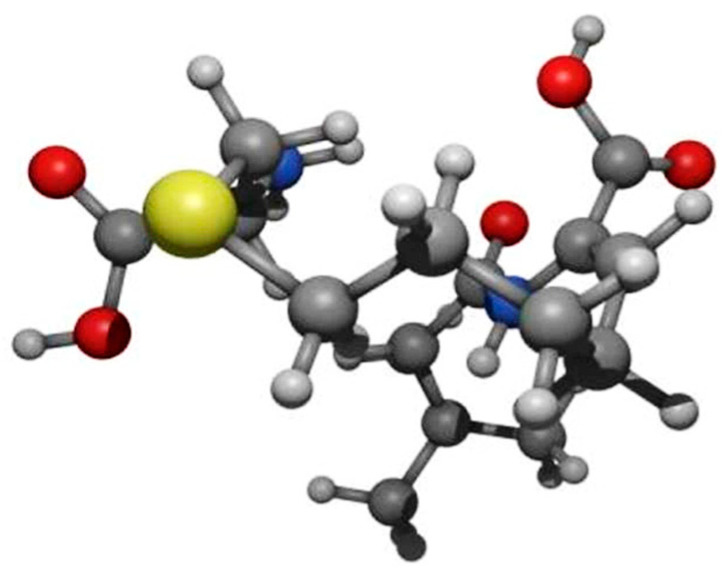
Chemical structure of the cilastatin molecule. Created by D. Ruben García, Arscreativa Studio (www.arscreativa.com (accessed on 31 July 2025)).

## Data Availability

The datasets used and/or analyzed during the current study are available from the corresponding author upon reasonable request.

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
