# Peer review of "Cilastatin Attenuates Acute Kidney Injury and Reduces Mortality in a Rat Model of Sepsis"

_ijms, 2025, doi:10.3390/ijms26167927_

Round 1

Reviewer 1 Report

Comments and Suggestions for Authors

The comment to the authors has been attached in the below document.

Author Response

Answer to Reviewer # 1

GENERAL COMMMENTS:

Question 1:

The reviewer states: “The writing of the Introduction was good but from the results part, the standard of scientific writing is below par which must be addressed, several words, terms and the way written must be thoroughly checked. As there are many mistakes, I recommend the authors to thoroughly correct them”.

Answer 1:

Many thanks to the reviewer for their review and appreciation. We apologize for any errors in the writing. We have now completely revised the wording of the article and proofread by a style editor to correct both style and language. The changes made can be seen in change control throughout the text.

Question 2:

The reviewer states: “Most of the Figures are very poorly made. No proper labelling. Surprisingly the molecular weights of the proteins in the WBs were provided in the supplementary, but nothing in the original figures. The aesthetics of the figures is of low quality. Even in the case of IF images, no proper labelling within the Figures were given. The quality some of the IHC Images are not up to the mark. The tissue regions seem not similar in all groups, no arrow mark indications in all images etc are a few of the things regarding the figures which the authors must improve”.

Answer 2:

The reviewer is right, and we greatly appreciate his honesty. We have greatly improved the figures so that the staining can be clearly seen, and we have appropriately marked with arrows, arrowheads, or asterisks the parts we want to highlight so that the reader can see them clearly. We have improved all the things that the reviewer mentioned and have changed the photographs of some figures to show a similar region for each photograph in each group. We have always tried to include a glomerulus in each of the photographs so that they can be compared with each other. We have tried to place them in the center, although this has not always been possible, but there is always a glomerulus for comparison.

The reviewer will be able to see the changes in the new images in the article, as well as in the figure captions.

Question 3:

The reviewer states: “In case of the material and methods please for all your submissions, please indicate accurate, proper and complete catalogue numbers for all reagents used, especially antibodies and chemicals. There are so many abs that every company has and for other researchers to reproduce, it is important to have exact catalogue numbers. I hope the authors will remember this for all future submissions”.

Answer 3:

The reviewer is indeed correct, and it is necessary to include references for antibodies and reagents so that other researchers can replicate our results.

We have now included this information in the Materials and Methods section.

Question 4:

The reviewer states: “In all the Figures, please replace the bar graphs with bar graphs with dots, which allows to visualize the deviation. Figure legends were written like a lab note book. They must be improved with proper n numbers indicated and required details of significance, stats etc, where needed”.

Answer 4:

As suggested by the reviewer, we have replaced the bar graphs with dot graphs and rewritten the figure captions accordingly. With this and the new improved images of the immunohistochemistry and Western blots, we have created new figures that the reviewer will find in the text.

MAJOR CONCERNS:

Question 1:

The reviewer states: “Fig 1D: Y-axis is mentioned as proteinuria, but here the authors just mentioned proteinuria. Consider clear labelling. In 1E, 1G, the authors can just add the protein as HE, KIM-1 etc, which will be easy for the readers immediately to know what is stained. 1H – what is Y-axis KIM-1 staining score, better to indicate exact term. Even for the 1G images a higher magnification can be provided where the KIM-1 is highlighted. Indicate in the legends also how many regions in each section have been quantified. In the legends the authors indicate brush borders, but for beginners to get the knowledge, please indicate clearly what authors want to reader to see. There are no visible arrows as mentioned by authors”.

Answer 1:

On the Y axis of Figure 1D, we have changed the name “Proteinuria” to “Urine proteins.

With respect to figures 1E and 1G, although the caption clearly details which stain corresponds to each image, we have added the abbreviated names of the stains as suggested by the reviewer, for greater clarity.

Regarding the legend for the Y axis in Figure 1H, KIM-1 staining (score), we would like to refer to the evaluation of KIM-1 staining in the kidney images. For better understanding, we have replaced it with “KIM-1 staining intensity (score).”

In addition, following the reviewer's instructions in Figure 1G, we have introduced the enlargement of a tubule (marked with a red asterisk inside) so that the staining can be clearly observed in CLP animals, nothing in control and control + cilastatin animals, and how the staining decreases greatly in CLP animals treated with cilastatin.

We have also indicated in the figure caption that the entire samples were quantified by analyzing the stained area, and we have clarified through symbols what the reader should see in the images.

Question 2:

The reviewer states: “The writing of the results part must be improved”.

Answer 2:

We have rewritten and corrected the results section as advised by the reviewer.

Question 3:

The reviewer states: “Line 129 - Values of kidney injury molecule-1 (KIM-1), Please indicate clearly. What values are the authors describing? There should be some description, at least brief, or indicate if it is written in methods part”.

Answer 3:

These values refer to the quantification of the stained area in the images of the samples for each animal. The quantification of immunohistochemistry is described in the materials and methods section (line 595 in the first version of the manuscript).

In any case, to clarify what the values are, we have rewritten the paragraph as follows: “Expression levels of kidney injury molecule-1 (KIM-1), a specific marker of kidney damage, obtained by quantifying the labeled area in the images of the samples for each experimental group, were significantly increased…”.

The reviewer can find this change on lines 166-168 of the new manuscript.

Question 4:

The reviewer states: “Line 143 - The authors can also indicate the name what is being referred to here as this is the first term used”.

Answer 4:

The reviewer is right. We have added the name and abbreviation. The changes can be found on line 184 of the new manuscript.

Question 5:

The reviewer states: “Fig- 2: The images are not sharp and convincing, especially CLP+Cil. Also, provide arrows in the CLP condition, so the readers can compare. Also provide similar areas in all conditions eg. With glomeruli in the middle. In case of ICAM and VCAM, the regions look very different. Even the MCP-1 staining is not at all convincing, even in the CLP condition. Satisfactory images are mandated to get to a conclusion about this Fig. There are very good abs that work also good in WB, which can be tried”.

Answer 5:

We have improved the figure by enhancing the quality of the photos to see the stains more clearly, and in some cases we have redone the figure. We have always tried to include a glomerulus in each photograph and to center it, although sometimes it is slightly off-center. The photographs are the same size, and we have tried to choose similar fields so that they can be compared with each other. Arrows have also been added to show the tubular staining, and arrowheads to show the staining in the glomerulus and bring the reader closer to what they should be seeing.

The reviewer will find these changes in the new Figure 2 of the manuscript.

Question 6:

The reviewer states: “Line 180: the arrival of leukocytes at the focus of – the sentence could be changed”.

Answer 6:

This sentence has been changed. The new sentence reads as follows: “Adhesion molecules such as vascular cell adhesion molecule-1 (VCAM-1) and intercellular adhesion molecule-1 (ICAM-1), which mediate leukocyte extravasation to inflamed tissues…”.

The reviewer can find the change in lines 250-252 of the new manuscript.

Question 7:

The reviewer states: “Line 187: Lines 146-148, the authors mention that Cilastatin significantly decreased the influx of monocytes/macrophages into the kidney, as well as levels of expression of MCP-1(Figure 2 A-C). Here in line 187-188 it was written that Cilastin alone did not affect MCP-1...”.

Answer 7:

Indeed, cilastatin significantly decreases the influx of monocytes/macrophages into the kidney, as well as MCP-1 expression levels in animals with CLP. The phrase “Cilastatin alone...” refers to the fact that when the control group is treated only with cilastatin (in this case there is no CLP, no damage), acting alone it has no effect on MCP-1 expression or on the recruitment of monocytes/macrophages to the kidney. As shown by both immunostaining and quantitative analysis, the levels in the cilastatin control group were comparable to those observed in the untreated control group.

To clarify this fact, we have changed the sentence and the new therefore reads: “Cilastatin alone had no effects on MCP-1 levels, monocyte/macrophage infiltration, or ICAM-1/VCAM-1 expression in non-septic animals compared with sham-operated control group”. The change can be found in lines 260-262 of the new manuscript.

Question 8:

The reviewer states: “Fig3: I haven’t seen use of VDAC-1 as a house keeping gene, "Please explain why VDAC and not ACTB, Gapdh, Tubulin etc. The quality of figure must be improved. Provide the ab molecular weights, add some borders etc. 3C: what is n-fold, with what house keeping gene is the normalization, the same VDAC-1?”.

Answer 8:

The reviewer is right to comment that it is not one of the most commonly used. We chose to use this protein as a loading control due to its high stability and consistent expression in tissues with a high mitochondrial density, such as the kidney. The kidney is a highly metabolic organ with a rich mitochondrial population, particularly in the proximal tubules. VDAC-1, being located in the outer mitochondrial membrane, is abundantly and stably expressed in renal tissue, even under various physiological and experimental conditions. Its relevance in our study is further supported by the fact that we analyzed several proteins involved in processes such as oxidative stress and apoptosis, which primarily affect mitochondria. Thus, VDAC-1 served as a more specific loading control than cytosolic proteins like GAPDH or β-actin. Another important reason for its selection is the lower variability of VDAC-1 under certain pathological conditions. Unlike actin or tubulin—whose expression can be altered in response to hypoxia, inflammation, fibrosis, or cell proliferation—VDAC-1 expression tends to remain stable, particularly in post-mitotic tissues such as the adult kidney. This was confirmed in our initial assessments, where we observed that actin and tubulin, as well as in other nephrotoxic and endotoxin-induced injury models previously developed by our group, were affected under damaging conditions, compromising their use as reliable loading controls.

Although it was not our primary objective, VDAC-1 also serves as a good marker of the mitochondrial fraction in total lysates. In mitochondria-rich tissues such as the kidney, VDAC-1 consistently reflects the mitochondrial content of the lysate and could be used as a normalizer in studies focused on mitochondrial function.

We acknowledge that this is not a conventional choice; however, it has proven to be a reliable control that has not presented any technical issues. Furthermore, many of the proteins studied in our model have molecular weights similar to those of actin and tubulin. Therefore, we opted for VDAC-1, which has a distinct molecular weight of approximately 30–35 kDa lower than those controls, minimizing potential band overlap or interference during antibody detection and allowing clear differentiation from our target proteins.

As suggested by the reviewer, we have improved figure 3, taking into account all of their recommendations. We have also added arrows or similar elements to focus the reader's attention on what we want them to see, and we have improved the quality of the figure in general and the images in particular.

Regarding Figure 3C, in the context of quantitative reverse transcription PCR (qRT-PCR), the term n-fold refers to the relative change in gene expression between two experimental conditions, typically calculated using the comparative Ct (ΔΔCt) method. It indicates how much more or less of a specific gene is present in one sample compared to another. An n-fold increase indicates how many times higher the expression of a target gene is in the test sample compared to the control simple (in this case, GAPDH, was our housekeeping gene, as described in the materials and methods section of the article, lines 938-941). Conversely, an n-fold decrease indicates reduced expression relative to the control. Thus, the n-fold value is a dimensionless ratio that allows for intuitive interpretation of upregulation or downregulation of gene expression. This is commonly used in relative quantification experiments to assess changes in gene expression.

Question 9:

The reviewer states: “3D: I would like to see some structural marker along with TLR4. The regions seems to be very different. Not convincing”.

Answer 9:

We can assure the reviewer that the regions shown are the same in size and shape for the different groups, since they are taken at the same time on the same day and all in the same way. If the reviewer looks at the figures, they will see how the tubules of similar sizes are drawn between the different images. Photographs are shown where the staining is correct and with significant differences. We did not include a structural marker in the renal tissue immunofluorescence analysis of TLR4, as its localization is well characterized in the literature, particularly in proximal tubular epithelial cells and glomerular endothelial cells. Additionally, we observed a marked difference in TLR4 expression in the sepsis groups compared to the control and treatment groups, which helps us validate the results found by Western blot. Given the established distribution of TLR4 in renal tissue under both physiological and pathological conditions, the use of an additional structural marker was considered unnecessary for the purpose of this study.

This decision is supported by previous studies demonstrating the constitutive and inducible expression of TLR4 in renal tubules and vasculature, particularly under inflammatory stimuli (e.g., Akira et al., 2006; Pulskens et al., 2008).

Question 10:

The reviewer states: “3I : Use similar regions for presenting the images. Control looks different to others. and try locating the glomeruli in the center. Like in the initial HE images, similar and glomeruli regions can be presented”.

Answer 10:

Figure 3I has been remodeled and improved to show similar regions and see the staining more clearly. All images have at least one glomerulus to make similar regions, where we have tried to center them as much as possible, although in this case the staining was much more abundant at the tubular level, as can be seen, and we did not want to place them exactly in the center in order to observe the tubular staining to a greater extent.

The glomerular staining in this case is not as dramatic as it was with the presence of hemorrhages in the hematoxylin/eosin stains. Here, if you enlarge the images, you can see that there is slightly more staining in the glomeruli of the CLP animals compared to the rest, but it is quite mild staining.

The major differences in staining are found in the tubular area.  Furthermore, the figure itself has several sub-figures and graphs, which would make it more crowded, and we do not believe that it provides essential information for the reader in this case.

Question 11:

The reviewer states: “Lines 228-230 The writing can be improved”.

Answer 11:

We have rewritten the sentence referred to by the reviewer.

Now the sentence reads as follows: “Nuclear factor kappa B (NF-κB), the downstream effector of TLR4 signaling, regulates numerous inflammatory mediators”.

The change can be found in lines 319-320 of the new manuscript.

Question 12:

The reviewer states: “Fig 4a: Once again not a great standard of Western blots. There are some excellent abs for Western blots. Why did the authors use NFkB instead of pNfKB? It is important to show pNFkB. It is important to show pNFkB and total NfKB. Most of the functions of NfkB are after phosphorylation and the authors totally ignored this fact and there is not even an indication about it. In the methods section, if there is a proper catalogue number, then atleast it would have been possible to check it. This can convey the mechanistic insights of the authors are not appropriate. Or do authors convey having investigated only total NfkB? VDAC1 seems to have been done on other gel. So, no housekeeping gene done on the same gel”.

Answer 12:

The reviewer is right, and this western is not one of the best we have ever done.  We have tried to improve it for better viewing. It is true that there are good antibodies, and this one in particular works very well for us in other models of kidney failure, but here it has not worked so well, and it has taken a little longer to obtain a clear band. In any case, upon closer examination of the western blots, it is possible that the upper band, which is slightly below 75 kDa, could also be p65. In fact, p65 should be closer to the 75 kDa marker than to the 50 kDa marker. However, we have decided to be as strict as possible, as we believe that the upper band is slightly elevated. Therefore, we have selected the lower band, as indicated in the following figure, which is available in the additional material sent with the western blot images. Although we cannot rule out that it is the upper band, the most important thing is that in both cases the result is the same, since there is more p65 in the CLP samples in both bands. The validation of the western data is provided by immunofluorescence, where again there is much higher fluorescence intensity in CLP animals.

                                                                                       ¿NFkβ?

With regard to the reviewer's statement that VDAC was not tested in the same gel, we must say that their assessment is incorrect, since both proteins were analyzed on the same SDS-PAGE, as the reviewer can analyze in the complete westerns as we have sent to the journal. The reviewer can verify this in the image above, which is the image after revealing the VDAC, and where we have both proteins at the same time, NFkβ and VDAC as housekeeping. In the following image, which is the same, we mark the position of VDAC:

The reviewer's doubts may be due to the smile shown by the VDAC-1 bands. However, differential migration between the gel regions was observed, with the low molecular weight bands exhibiting a slight "smiling" effect, while the high molecular weight region showed more uniform migration. This common artifact was taken into account during analysis and does not affect the interpretation of the results.

On the other hand, in this study, we analyzed total NF-κB i.e., the total amount of NF-κB protein present in a cell, regardless of its activation state. The molecular mechanisms underlying tissue damage are well characterized, and it is well established that NF-κB increases its expression and is activated and plays a key role in initiating the inflammasome response. Unfortunately, it is not feasible to analyze all individual proteins involved in the damage pathways; therefore, we focused on the most relevant targets, those we consider most informative for the objectives of this study. While phosphorylated NF-κB provides specific insights into its activation status, total NF-κB still offers valuable information regarding the involvement of this signaling pathway. A more detailed approach—such as analyzing subcellular fractions and comparing cytoplasmic and nuclear localization—would indeed warrant the separate evaluation of phosphorylated and non-phosphorylated forms. However, under our experimental conditions and to the best of our knowledge, total NF-κB expression was sufficient to reflect the activity of this key regulatory pathway and to support our conclusions regarding inflammasome activation.

As recommended by the reviewer, we have added the catalog and reference numbers of the products used in the experiments to the materials and methods section.

Question 13:

The reviewer states: “Fig 5c: I am sorry, the Casp-1 ab western is below the standard of presentation for a publication. There are excellent abs for these proteins. The authors even in this case show only Caspase-1, is it cleaved form or what? Even in the next blot only a Pro-IL1b band. Showing no other forms fails to convince the hypothesis of the authors. The corresponding quantifications are hard to accept with such blots. I am sorry but it is a fact that the standard of the Western blots is below par for publication”.

Answer 13:

The reviewer is right about the image quality of the caspase 1 western blot. We have tried to improve it, and we believe that the bands are now clearer. In any case, the gel ran and stained perfectly, as the reviewer can see in the western blot images uploaded to the platform and in the image shown here. The antibody is good, but the problem is that the caspase 1 band, which is procaspase-1 and appears at around 45 KDa, is very thin, as can be seen, and when it is cut and enlarged for the article, it loses some of its quality and sharpness. However, we have improved it, and the reviewer can find it in the new Figure 5.

Furthermore, we have already mentioned that this band corresponds to pro-caspase-1, but following the reviewer's instructions and studying the Abcam datasheet, the antibody also recognizes the activated form, producing a band between 10-20 KDa. Although also weakly stained, we have included the result of the activated form of caspase. In both cases, both procaspase and the activated form are increased in CLP animals compared to controls, and their levels are reduced when cilastatin is administered to septic animals.

The determination of Caspase-1 (mature and activated form) is appropriate to support the involvement of the inflammasome pathway, a conclusion further supported by RT-PCR results and the expression of other components of the inflammasome complex.

Regarding pro-IL-1β, we consider its selection to be appropriate, as it represents the precursor of the mature cytokine and provides relevant information comparable to that of the active form. Moreover, due to its higher molecular weight, pro-IL-1β is more amenable to detection and quantification in immunoblotting assays. The validity of this approach is further supported by the RT-PCR data. Importantly, the detection of pro-IL-1β complements our findings of activated caspase-1 and increased NLRP3 expression, confirming activation of the NLRP3 inflammasome pathway. Given that active caspase-1 is responsible for the cleavage of pro-IL-1β into its mature, bioactive form, the simultaneous detection of these components supports the presence of a functional inflammasome response in our model.

Question 14:

The reviewer states: “Figure A: Please see excellent Sirius red staining. The imaged are blurred and not a regular kind of color of Sirius red that I know. Not convincing et al”.

Answer 14:

The reviewer is right that some of the images of the Syrian red were blurred, but the color is basically what we got when we used the technique. We have now modified the figure and kept the initial colors exactly as we took the figure, without applying contrasts or anything else. Now all the photos are displayed correctly and none are blurry. The color, as I said, is the color of the technique itself.  Depending on the tissue being stained, a more yellowish (or reddish background color may appear, with a strong red indicating collagen. In our case Sirius Red staining was performed under standardized conditions. Nevertheless, it is important to acknowledge that this technique can be influenced by several variables depend of the protocol used. Differences in tissue fixation, dehydration and embedding procedures can affect collagen fiber integrity and dye penetration. Additionally, variations in section thickness and timing of staining and washing steps may alter the intensity and specificity of the signal. The type of tissue to be stained also has an influence, as it can give a more yellowish or reddish/pinkish background, as was the case with our kidney samples.  In any case, other authors performing the technique on kidneys have obtained results very similar to those found in our study in terms of color, such as:

-Quantification and Comparison of Anti-Fibrotic Therapies by Polarized SRM and SHG-Based Morphometry in Rat UUO Model. PLOS One 201611(6): e0156734. DOI: 10.1371/journal.pone.0156734

-Zinc deficiency during growth: Influence on renal function and morphology. Life Sciences 2007, 80(14):1292-302. DOI: 10.1016/j.lfs.2006.12.035.

We encourage the reviewer to look at them to see how the staining is completely similar to that found in our results. In any case, however, all staining runs were performed in parallel and imaging conditions were kept constant, including microscope settings and exposure times, to minimize technical variability.

Question 15:

The reviewer states: “How did the authors use only mice for the survival analysis? It is too less in my opinion. Did the authors perform any apriori analysis? Minimum n=10 is important for such analysis”.

Answer 15:

In our experiment, we used rats to conduct a proof of concept survival test for up to 48 hours, using an aggressive CLP model, during which time we knew that mortality was practically 100% due to the massive infection in these animals. What we wanted to observe here is that renal protection with cilastatin would be capable of prolonging the life of animals with sepsis, confirming the well-known clinical evidence that in patients with sepsis and acute renal failure, mortality is about 75%, while in those with sepsis without AKI, mortality ranges between 27% and 32% (Liétor A, et al. NefroPlus. 2010; 3(3):9-19, doi:10.3265/NefroPlus.pre2010.Nov.10733; Poston JT, et al.  BMJ. 2019;364:k4891. doi: 10.1136/bmj.k4891; Zhang CF, et al. Eur Rev Med Pharmacol Sci. 2020; 24(10): 5604-5617. doi: 10.26355/eurrev_202005_21346). We wanted to conduct a pilot study to verify whether the renal protection provided by cilastatin could prolong the animals' lives, which would have enormous translational implications for human medicine. And we have achieved our goal.

The group has proven experience in performing this animal model and is knowledgeable about the time of death by modulating the amount of extruded fecal content, which can alter the severity of the model. We already knew that with this aggressive model, animals tend to die within 48 hours. However, for the rest of the study, by performing a less aggressive CLP, we were also aware that 100% of the animals would be alive after 48 hours, with some of them beginning to die after that point.

Therefore, six animals per group was sufficient to demonstrate the protective efficacy of cilastatin in this preliminary approach, and in this way we did not use as many animals, complying with the 3Rs rule in animal experimentation and specifically the “reduce” principle, which focuses on decreasing the number of animals used in experiments while maintaining the scientific validity of the results. In this case, with n = 6 animals per group, we see that mortality decreased by approximately 30%, accurately reflecting the results found at the clinical level. Therefore, we can conclude that under these more extreme conditions, cilastatin was shown to significantly reduce animal mortality, effect that will be partly due to the renal protection exerted.

Final comment

All these are important points which must be addressed by the authors for a better manuscript preparation. I hope my comments will be helpful for a better manuscript preparation and future submission

Many thanks to the reviewer for their comments and review, as they have greatly contributed to improving the article.  Thank you.

Reviewer 2 Report

Comments and Suggestions for Authors

Comments

The manuscript with entitled “ Cilastatin attenuates acute kidney injury and reduces mortality in a rat model of sepsis”.

  1. This manuscript discusses the anticancer activity of Cilastatin. This topic is highly relevant and interest in health. It is intriguing. The manuscript is original.

  1. In the introduction, why sepsis can cause acute kidney injury? The author should introduce the chemical structure and molecular formula of Cilastatin. Recommendation: The author to provide additional clarification.

  1. In the methodology, What are the characteristic signs of the model of sepsis-induced kidney injury? How does the author determine the success of the model? How to judge the success of model establishment? Are Cilastatin clinically available? If it is used clinically available, what is the dose used? What is the relationship between the dose of clinically and the dose used in this study? What is base of the dose of Cilastatin used in this study?

  1. In the conclusions, the clarity of the image of Western blot should be improved. What are the specific limitations of the article and future research directions? There a multiple grammar and style errors, wrongly constructed phrases and scientifically incorrect expressions.

  1. The format of references is consistent, and it meets the requirements of the journal.

Author Response

Answer to Reviewer # 2

Question 1:

The reviewer states: “This manuscript discusses the anticancer activity of Cilastatin. This topic is highly relevant and interest in health. It is intriguing. The manuscript is original”.

Answer 1

Many thanks to the reviewer for their comments. Acute kidney injury associated with sepsis is a serious problem with a mortality rate ranging between 50% and 80% in the intensive care units, and there is currently no drug that can be used to protect the kidneys from damage. However, it is well known that in patients with sepsis and acute renal failure, mortality is about 75%, while in those with sepsis without AKI, mortality ranges between 27% and 32%. That is why it is very important to protect the kidneys in sepsis. The manuscript discusses the therapeutic power of cilastatin in preventing acute renal failure induced by polymicrobial sepsis.

Question 2:

The reviewer states: “In the introduction, why sepsis can cause acute kidney injury? The author should introduce the chemical structure and molecular formula of Cilastatin. Recommendation: The author to provide additional clarification”.

Answer 2

It is well documented that sepsis is the main cause of acute kidney injury (AKI), typically emerging within the early hours of its onset, with 30–60% of septic patients developing AKI. This fact is well reflected in some of the references cited in the article, such as numbers 4, 5 and 7. Particularly, a very recent one, number 9, reviews perfectly and extensively the fact of how sepsis is able to generate AKI, and it is perfectly summarized in figure 1 (Acute Kidney Injury in Sepsis. Int J Mol Sci 2024, 25, 5924. DOI: 10.3390/ijms25115924). In addition, sepsis is the leading cause of AKI in critically ill patients, accounting for 40-70% of cases. And the main problem is that patients with AKI already have a higher risk of death than other critically ill patients, as it is an independent risk factor associated with increased mortality (Regueira T, et al. Med Intensiva. 2011;35(7):424-32, doi: 10.1016/j.medin.2011.03.011).

Briefly, sepsis can lead to AKI due to a combination of pathophysiological factors that compromise blood flow (renal perfusion) and renal function that lead to a rapid decline in glomerular filtration rate (GFR). Recent studies have shown that inflammation and apoptosis are the primary pathological processes in septic AKI. This has been demonstrated through the progression of organ damage, including both macrovascular and microvascular dysfunction (systemic hypotension and renal vasoconstriction, not necessarily within the ischemic range), immune dysregulation and alterations in oxygen delivery, either due to decreased perfusion or impaired diffusion secondary to edema and inflammation. The kidneys are particularly susceptible to the septic environment and this vulnerability is considered a key factor contributing to the development of AKI. The kidney plays a critical role in filtering blood and is constantly exposed to circulating inflammatory mediators, pathogen-associated molecular patterns (PAMPs), and damage-associated molecular patterns (DAMPs), all of which activate immune responses within renal tissue. Tubular epithelial cells and resident immune cells express a wide range of pattern recognition receptors (PRRs), such as Toll-like receptors, which amplify the inflammatory cascade. This results in increased oxidative stress, mitochondrial injury, and apoptosis, contributing further to renal dysfunction. The limited regenerative capacity of renal tubular cells in the context of ongoing inflammatory insult makes the kidney especially prone to sustained injury in sepsis.

In addition, the presence of bacterial products in the systemic circulation activates inflammatory cells that infiltrate renal tissue and increases the secretion of cytokines such as TNF-α, IL-1β, and IL-6 by tubular epithelial cells, as well as leukocyte activity. Intraglomerular thrombosis and intratubular obstruction may occur, as fibrin deposits have been observed within the lumens of glomerular capillaries. Medullary tubular epithelial cells show vacuolization and increased expression of Kidney Injury Molecule-1 (KIM-1), a marker of tubular injury. Mitochondrial dysfunction and oxidative stress are also present, with increased production of reactive oxygen species (ROS) and induction of nitric oxide synthase (NOS), which can lead to damage of the endothelial barrier and glycocalyx.  All these structural and functional changes contribute to the tubular dysfunction characteristic of sepsis-induced AKI.

Despite receiving approximately 20–25% of the cardiac output, the renal microcirculation is tightly regulated and highly heterogeneous, with distinct perfusion patterns between the cortex and medulla. This makes the kidney especially susceptible to even subtle alterations in systemic hemodynamics and local oxygen delivery. During sepsis, systemic inflammation, endothelial dysfunction, and microvascular dysregulation impair renal autoregulation and lead to regional ischemia, particularly in the outer medulla, which is already at the borderline of hypoxia under normal conditions.

As a result, AKI represents a serious complication of sepsis, and the consecuence is that patients with AKI already have a higher risk of death than other critically ill patients, as it is an independent risk factor associated with increased mortality (Regueira T, et al. Med Intensiva. 2011;35(7):424-32, doi: 10.1016/j.medin.2011.03.011). A patient with sepsis complicated with AKI has a significant increase in relative mortality (70%) with respect to another without AKI (Liétor A, et al. NefroPlus. 2010; 3(3):9-19, doi:10.3265/NefroPlus.pre2010.Nov.10733; Poston JT, et al.  BMJ. 2019;364:k4891. doi: 10.1136/bmj.k4891; Zhang CF, et al. Eur Rev Med Pharmacol Sci. 2020; 24(10): 5604-5617. doi: 10.26355/eurrev_202005_21346).

We have tried to summarize all this in a few lines in the introduction of the article, and specifically the reviewer can find it in lines: 57-63 of the new manuscript.

In response to the second question, cilastatin is the monosodium salt of [R-[R, S-(Z)]]-7-[(2-amino-2-carboxyethyl)thio]-2-[[(2,2-dimethylcyclopropyl)carbonyl]amino]-2-heptenoic acid. Its empirical molecular formula is C₁₆H₂₅N₂NaO₅S, with a molecular weight of 280.44 Da. These data, together with the chemical structural formula of cilastatin, which is the new Figure 9, have been incorporated into the article in the Materials and methods section. The reviewer can find them between lines 782 to 796 of the new manuscript.

Question 3:

The reviewer states: “In the methodology, What are the characteristic signs of the model of sepsis-induced kidney injury? How does the author determine the success of the model? How to judge the success of model establishment? Are Cilastatin clinically available? If it is used clinically available, what is the dose used? What is the relationship between the dose of clinically and the dose used in this study? What is base of the dose of Cilastatin used in this study?”.

Answer 3:

Answer to the question: “In the methodology, What are the characteristic signs of the model of sepsis-induced kidney injury?”

The cecal ligation and puncture (CLP) model has been widely used in the last 30 years to study the pathophysiology of sepsis and is well described in the literature (González-Nicolás MA & Lázaro A). Compared to other models, it provides a better representation of the complexity of human sepsis where immune, biochemical, and hemodynamic responses are similar to those observed in humans (J Biomed Sci. 2017; 24(1):60. doi: 10.1186/s12929-017-0370-8). Not in vain, it has been considered the "Gold standard" for showing a high degree of similarity to the progression of human sepsis and tends to be the technique of choice for developing models of sepsis-induced acute kidney injury (AKI) (Trends Microbiol. 2011; 19(4):198-208. doi: 10.1016/j.tim.2011.01.001; González-Nicolas MA & Lazaro A).

Sepsis-induced AKI presents with a distinct set of histological, molecular, and functional features that differentiate it from other forms of renal injury. Unlike ischemic or nephrotoxic models, where widespread tubular necrosis is common, septic AKI is often characterized by minimal cell death and instead displays sublethal injury, including tubular epithelial cell dysfunction, loss of polarity, and cytoskeletal disorganization. One hallmark of this model is the presence of inflammatory cell infiltration, especially neutrophils and macrophages, in the interstitial space, driven by systemic and local release of pro-inflammatory cytokines such as TNF-α, IL-6, and IL-1β.

Another characteristic feature is endothelial dysfunction, manifesting as loss of barrier integrity, shedding of the glycocalyx, and impaired microcirculatory perfusion. These alterations contribute to heterogeneity in renal blood flow and focal hypoxia, particularly in the outer medulla. Mitochondrial injury is also prominent, with reduced ATP production, increased generation of reactive oxygen species (ROS), and activation of cell stress pathways such as the unfolded protein response and mitophagy.

Additionally, upregulation of biomarkers such as Kidney Injury Molecule-1 (KIM-1), Neutrophil Gelatinase-Associated Lipocalin (NGAL), and increased expression of inducible nitric oxide synthase (iNOS) are molecular indicators of injury in experimental models. Functional signs include an abrupt decline in glomerular filtration rate (GFR) without necessarily corresponding to marked tubular necrosis, supporting the notion that sepsis-associated AKI is primarily driven by inflammation, microvascular alterations, and metabolic reprogramming rather than classical ischemia.

Answer to the questions: “How does the author determine the success of the model? How to judge the success of model establishment?”

The CLP model was developed at the Getafe Hospital and was carried out by researchers from Dr. Jose Angel Lorente's group, who have a fully accredited experience in the realization of this model and others of septic damage in small and large animals. Its highly trained staff (Mario Arenillas, the author of the article is the veterinarian responsible for the animal facility), knows it and performs it to perfection, based on the greater or lesser aggressiveness required, as has been the case in the work presented here where two models have been made, the second one much more aggressive and where the animals died before 48 hours. It is a model that they have made many times and in which they have determined and studied many parameters, mainly with respect to lung damage and renal damage. And this is confirmed by some of the publications where they have made this same model precisely for studies of lung and kidney damage.  Some of them are:

Seija M, et al. Role of peroxynitrite in sepsis-induced acute kidney injury in an experimental model of sepsis in rats. Shock. 2012; 38(4):403-10. doi: 10.1097/SHK.0b013e31826660f2.

Izquierdo-García JL, et al. A metabolomic approach for diagnosis of experimental sepsis.Intensive Care Med. 2011; 37(12):2023-32. doi: 10.1007/s00134-011-2359-1.

Nicolás Nin, et al. Vascular dysfunction in sepsis: effects of the peroxynitrite decomposition catalyst MnTMPyP. Shock. 2011; 36(2):156-61. doi: 10.1097/SHK.0b013e31821e50de.

Chacon-Cabrera A, et al. Influence of mechanical ventilation and sepsis on redox balance in diaphragm, myocardium, limb muscles, and lungs. Transl Res. 2014; 164(6):477-95. doi: 10.1016/j.trsl.2014.07.003.

From this accredited experience in its realization, we know that the model is well done, and its achievement was successful because the results that we saw later regarding renal damage and that we present in this article, validate the success of the model and the acute renal failure that we pursued with its realization. But, before the slaughter of the animals, clinical signs indicative of systemic deterioration were evident of a correct development of the model, including anorexia and lethargy, abnormal gait or a hunched posture, persistent curled-up position, alterations in grooming behavior, piloerection, vocalization upon handling suggestive of pain or discomfort, partially closed eyes, and changes in fecal and urinary output. These manifestations are consistent with established clinical scoring systems used to assess the severity of illness and to ensure animal welfare in experimental sepsis models.

On the other hand, in sepsis, a hypodynamic or immunosuppressive phase occurs characterized by a profound anti-inflammatory response involving reduced lymphocyte proliferation and function, as well as increased apoptosis. In an unpublished hematological study, we observed that the CLP group exhibited a significant reduction in total leukocyte and platelet counts. Neutrophil percentages increased threefold compared to control values, while monocyte levels doubled. Conversely, lymphocyte and eosinophil percentages were significantly decreased. Notably, we observed an increase in LUCs (Large Unstained Cells), which represent a small fraction of cells (<6%) that do not take up stain during differential leukocyte counts. These cells typically include hyperactivated large leukocytes, blasts, atypical large lymphocytes, plasmocytes, and other pathological cells. An elevated LUC count (>6%), especially when accompanied by increased transaminase levels, is considered indicative of systemic infection or ongoing inflammation. All hematological parameters significantly improved in the cilastatin-treated group, showing a reduction in neutrophil and macrophage infiltration along with restoration of lymphocyte levels. Red blood cell indices remained unchanged across all experimental groups.

In addition to this, we found clear results of renal failure, with elevated creatine and urea levels, and a clear decrease in glomerular filtration, data that indicated that sepsis had indeed caused renal failure. The presence of protein in the urine also informed us of glomerular damage, and the presence of the biomarker KIM-1 that only appears when there is tubular damage were key aspects to affirm that the model was well established and successful in its development. And as we have commented, cilastatin was able to prevent inflammatory damage by decreasing each characteristic mentioned.

Answer to the question:Are Cilastatin clinically available?

The direct answer is no, it is not possible to use cilastatin alone for human clinical use.  But it does exist in combination with an antibiotic imipenem to form the antibiotic imipenem/cilastatin (which has different trade names Primaxin, Tienam, Zienam...) which has been used in human clinical use since 1985. Therefore, cilastatin can only be used in the clinic in combination with imipenem. Its function is to inhibit the renal dehydropeptidase I enzyme so that it cannot metabolize and inactivate imipenem, preventing its absorption, increasing its urinary excretion, and reducing its concentration inside the tubular cell.

Our research group observed at the preclinical level that using cilastatin alone (without imipenem) as a therapeutic treatment, it was able to protect the kidney of rodents against toxic acute renal failure, produced for nephrotoxic drugs, which led us to think that other situations that induce renal failure, such as sepsis, could also benefit, which is the work presented here. Our results led us to patent the nephroprotective effect of cilastatin and to stablish the spin off Telara Pharma, with the aim of bringing cilastatin alone to clinical use as a nephroprotective drug.

Answer to the question:If it is used clinically available, what is the dose used?”

As mentioned above, cilastatin is not clinically available as a standalone drug, but it is used in combination with the antibiotic imipenem in the imipenem/cilastatin (I/C) formulation. A 1:1 ratio of I/C was established as the optimal dose to maintain inhibition for an interval of 8 to 10 hours (Norrby, S. R. et al. Antimicrob. Agents Chemother. 1983, 23, 300-307).

The dose of I/C used depends on the severity of the infection. In adults and adolescents, the usual doses are 250 mg/250 mg for mild infections every 6 hours (1 gram total per day), 500 mg/500 mg for moderate infections every 8 hours (1.5 grams total) or severe infections by very sensitive germs every 6 hours (2 grams total) and finally 1000 mg/1000 mg for severe and/or life-threatening infections every 6 or 8 hours (3- or 4-grams maximum total). The usual dose in children one year of age or older is 15/15 or 25/25 mg/kg/dose every 6 hours.

The data were obtained from the package inserts and technical data sheets of the drug imipenem/cilastatin.

Answer to the questions: “What is the relationship between the dose of clinically and the dose used in this study? What is base of the dose of Cilastatin used in this study?”.

The dose employed in this study was 150 mg/kg/day, determined based on prior experience of our research group, which has consistently yielded positive outcomes and consistent therapeutic efficacy as supported by our published studies on acute kidney injury induced by nephrotoxics such as cisplatin or gentamicin (references in the manuscript: 16, 17, 19, 23). But when we started our studies on this subject (Kidney Int. 2012 Sep;82(6):652-63. doi: 10.1038/ki.2012.199) we chose to administer that dose based on the literature we had available in this regard, which were old studies in which we administered imipenem/cilastatin and saw protection over nephrotoxic drugs, mainly cyclosporine and vancomycin, assessing functional values such as serum creatinine and urea. Specifically, Hammer C et al. in their article “Reduction of cyclosporin (CSA) nephrotoxicity by imipenem/cilastatin after kidney transplantation in rats” Transplant Proc. 1989 Feb;21(1 Pt 1):931, used 150 mg/kg/day bw showing nephroprotection; Nakamura T et al. in their articles: “Effects of fosfomycin and imipenem/cilastatin on nephrotoxicity and renal excretion of vancomycin in rats” Pharm Res. 1998 May;15(5):734-8. doi: 10.1023/a:1011971019868 and “Effects of fosfomycin and imipenem-cilastatin on the nephrotoxicity of vancomycin and cisplatin in rats”, J Pharm Pharmacol. 1999 Feb;51(2):227-32. doi: 10.1211/0022357991772187, also used a dose of 150 mg/kg/day bw and found protection against vancomycin nephrotoxicity in both studies; Sido et al. used 150 mg/kg day bw but twice a day, demonstrating its efficacy in the paper “Nephroprotective effect of imipenem/cilastatin in reducing cyclosporine toxicity” Transplant Proc. 1987 Feb;19(1 Pt 2):1755-8.  Finally, Toyoguchi et al. in rabbits tested various doses of cilastatin for the treatment of vancomycin nephrotoxicity, 75, 150 and 300 mg/kg/day bw, showing a dose-dependent effect on protection, but no major differences in renal protection between the 150 and 300 mg/kg dose.

Based on all these studies that had shown renal protection against cyclosporine and vancomycin, mainly at a dose of 150 mg/kg, we decided to use this same dose, but only cilastatin (without imipenem) in our protection models. We were able to verify how this dose was very effective in nephroprotection against toxins (cisplatin and gentamicin). We also performed experiments with doses of 75 mg/kg/day and 300 mg/kg/day administered to rats treated with cisplatin. The 75 mg/kg dose was shown to be less effective in nephroprotection, while the 300 mg/kg dose was shown to be slightly more effective than the 150 mg/kg dose but without significant differences, as described by Toyoguchi in rabbits for the treatment of vancomycin nephrotoxicity.

Based on all these studies, and on our own results in animal models of acute renal failure, 150mg/kg/day was the dose chosen for these studies, and we have maintained it for the study presented here, and it has also proved to be effective. We also knew that 150 mg/kg is a clinically relevant dose, as we had previously conducted an interspecies scaling using body surface area (BSA) normalization. According to FDA guidelines, the human equivalent dose (HED) of 150 mg/kg in rats corresponds to approximately 24.3 mg/kg in humans. For an average adult (70 kg), this translates to a total dose of ~1.7 grams, which is within the range of the maximum clinical dosage currently approved for cilastatin (1 gram every 8 hours, or up to 3 grams/day (or 4 maximaum) when co-administered with imipenem). Therefore, the experimental dose employed in our study is pharmacologically relevant and translationally appropriate for modeling potential clinical effects in the context of severe systemic inflammation and sepsis.

Question 4:

The reviewer states: “In the conclusions, the clarity of the image of Western blot should be improved. What are the specific limitations of the article and future research directions? There a multiple grammar and style errors, wrongly constructed phrases and scientifically incorrect expressions”.

Answer 4:

We have improved the images of the western blots for better and clearer visualization.

Our study, like all scientific research, has and has had some limitations, as well as methodological considerations, since the study of the pathophysiology of sepsis-associated acute kidney injury (AKI) faces several important limitations. One of the most significant is the lack of detailed information on renal histopathological changes in humans, as kidney biopsies are generally contraindicated and technically challenging in critically ill patients in the ICU setting. Beyond the clinical constraints imposed by sepsis itself, fully replicating the clinical syndrome under experimental conditions is also difficult. Available experimental models do not always accurately reproduce the complexity of septic syndrome. Nevertheless, the cecal ligation and puncture (CLP) model is widely considered the “gold standard” and is extensively described in the literature as one of the most frequently employed methods for this purpose.

Another challenge is the difficulty in obtaining reliable measurements of hemodynamic parameters in small animal models. Moreover, some rodent species have shown intrinsic resistance to endotoxins, which can limit the translational relevance of the findings. These limitations are compounded by the relatively short duration of experimental models compared to the often prolonged and clinically complex course of sepsis in human survivors.

One disadvantage of the CLP technique is the inability to precisely control the amount of fecal material extruded during cecal compression. Furthermore, intestinal microbiota composition is not uniform among individual animals or species, making inter-study comparisons challenging and requiring careful interpretation. Variations in surgical procedures and postoperative care also contribute to inconsistencies,    Standardization of the model is inherently difficult and largely depends on the technique employed and the surgeon’s level of experience. Additionally, the use of opioids such as buprenorphine may suppress respiration and locomotor activity, which can be misinterpreted as clinical signs of sepsis.

Another important limitation of our study was the identification of microbial species involved in the development of sepsis in the polymicrobial CLP model. These species may vary depending on the strain and species of the animal used, and this can also lead to significant differences between similar studies. In our case, we encountered logistical limitations that prevented a comprehensive analysis and characterization of the specific microbial populations contributing to the septic process.

The severity of the model has a direct impact on survival rates, as illustrated in Figure 7. Parameters such as the length of the ligated cecum, catheter gauge, and number of punctures can be adjusted to modulate the degree of severity and mortality. When performing the CLP model, all of these factors and technical parameters must be carefully considered to ensure reproducibility and consistency of results. Therefore, a thorough understanding and mastery of the procedure are essential to accurately implement the CLP model. In our experiment, we believe that this was not really a limitation, since, as we mentioned previously, the research team has extensive experience in implementing this model, although it is true that variations cannot be ruled out 100%, and the comparison in terms of damage may differ from other studies using the same model.

Additional limitations relate to space constraints associated with the use of metabolic cages, not only in terms of their physical placement but also regarding the number of units that can be managed simultaneously in a given experiment. The implementation of the 3Rs (Replacement, Reduction, and Refinement) further necessitates careful planning to minimize the number of animals used without compromising scientific validity. This was, for example, one of the reasons why we did not use a larger n in the 48-hour survival experiment (Figure 7).

Some of these limitations of the study have been incorporated into the manuscript, and the reviewer can find them in the discussion, between lines 758-767

With regard to future directions, this article highlights that cilastatin appears to be a universal nephroprotector for any type of acute renal failure of any etiology. We had already demonstrated the protective role in nephrotoxic-induced AKIs, and now we have done so in septic damage, showing that the pathways of renal protection by cilastatin are common to any type of renal damage.

This work opens the door to the study of other non-toxic acute renal failures, such as ischemia-reperfusion, or even kidney transplantation, where cilastatin may be beneficial for maintaining the donated kidney in cardiac arrest by administering it before reperfusion in the recipient. -Omic studies could also bring us closer to understanding the protective mechanism and the exact molecular pathways through which cilastatin exerts its nephroprotective action. On the other hand, this work highlights the beneficial effects of cilastatin and opens the door to future clinical use in this context, work that is being developed by the spin-off Telara Pharma (www.telarapharma.com)to bring the drug cilastatin to human clinical practice as a nephroprotective agent.

The text has been extensively revised and proofread by a style editor to correct both style and language. The changes made can be seen in change control throughout the text.

Question 5:

The reviewer states: “The format of references is consistent, and it meets the requirements of the journal”.

Answer 5:

Thank you very much for the review and appreciation.

Round 2

Reviewer 1 Report

Comments and Suggestions for Authors

Peer review_ijms-3734809_Revision-2

I thank the authors for their resubmission of the manuscript and applying my important comments. Please find the comments below about your corrections.

  • I still see the graphs not in a good resolution (the journal might ask for a better resolution images). Please check. The images look far better now.
  • Fig-4: With respect to the NFKB, the authors explanation is fine, and I would recommend to show both the bands, but the presented Ab especially for total should be far better. If possible try some Femto substrate.
  • Fig-5, Even here, atleast for the purpose of presentation the bands should be slightly visible, I am not asking for a think saturated one, but if you get these signals for 1 min, then why not expose the gel for 5 min or 10 min so that the signal is clear to quantify and visualize. It is necessary sometimes to do that. And the effect will not vanish with this. I would like to see better blots please. I am sorry these must be improved. If I am right, there is adjustment of brightness of the total blot, the blots can be exposed a bit. Normally 3-4 different exposed images are taken. Do the authors convey that these are long exposed bands? It is not for the reviewers- imagine if I see such a blot when reading and I know something is not great about it (let that be ab, exp, even who accepted those blots for the paper). I hope the experienced authors will understand the concern. For some rare or bad antibodies etc, I totally can accept them, but for such well established markers, such light bands are not convincing. I am satisfied with the explanation of the authors about using VDAC-1 as HKG, but see many blots the signal is so light, and it is also regulated in some blots (eg: 5c). Anyway, I hope the authors will address my concerns.

Thank you

Author Response

Answer to Reviewer # 1

Thanks to the reviewer for all their valuable insights.

Question 1:

The reviewer states: “I still see the graphs not in a good resolution (the journal might ask for a better resolution images). Please check. The images look far better now”.

Answer 1:

The figures were saved at a quality of 300 dpi, as recommended by the journal. They were initially created in PowerPoint, including the immunohistochemistry images we acquired using a photographic program attached to the microscope, the immunofluorescence images from the confocal microscope, and the western blot images saved directly from the developing device. They were then saved as TIFF files and saved from Photoshop at a resolution of 300 dpi. The process has been the same for our other publications, including other publications in MDPI (including the International Journal of Molecular Sciences), and we have never had any problems at this level.

In any case, to try to improve the resolution, we have saved them as jpegs at a new quality of 300 dpi, where we have also removed noise and tried to focus the images more for better resolution. They have then been saved at maximum quality. We hope that the reviewer will now see them with higher quality and resolution.

Question 2:

The reviewer states: “Fig-4: With respect to the NFKB, the authors explanation is fine, and I would recommend to show both the bands, but the presented Ab especially for total should be far better. If possible try some Femto substrate”.

Answer 2:

We understand the reviewer's request. Performing a western blot of NKFB and its activated form through phosphorylation is not a simple technique in the sense that they do not come out as clean and neat as other proteins. This is something we have verified on other occasions, which is why we previously used radioactive EMSAS, which were much more effective in terms of measurement, but had the problem of radioactivity and are no longer permitted for use in many places. In the laboratory, we do not have a single antibody for each protein we measure, but rather we have several, since depending on the models, some work better than others. We have repeated the NFKB western with several of them, and finally the one that worked best was the RELA/NFκB p65 (F-6): sc-8008, Santa Cruz Biotechnology.

We also performed a western blot of the phosphorylated protein, and although it was difficult to obtain because the images are not very good, we can see that it follows the correct pattern, with NFKB being much more activated in the kidneys of the animals in the CLP group.

The reviewer can see the images in the new Figure 4 of the article, where we have improved the quality of the total NFKB western blots and also added the phosphorylated protein.

Question 3:

The reviewer states: “Fig-5, Even here, at least for the purpose of presentation the bands should be slightly visible, I am not asking for a think saturated one, but if you get these signals for 1 min, then why not expose the gel for 5 min or 10 min so that the signal is clear to quantify and visualize. It is necessary sometimes to do that. And the effect will not vanish with this. I would like to see better blots please. I am sorry these must be improved. If I am right, there is adjustment of brightness of the total blot, the blots can be exposed a bit. Normally 3-4 different exposed images are taken. Do the authors convey that these are long exposed bands? It is not for the reviewers- imagine if I see such a blot when reading and I know something is not great about it (let that be ab, exp, even who accepted those blots for the paper). I hope the experienced authors will understand the concern. For some rare or bad antibodies etc, I totally can accept them, but for such well established markers, such light bands are not convincing. I am satisfied with the explanation of the authors about using VDAC-1 as HKG, but see many blots the signal is so light, and it is also regulated in some blots (eg: 5c). Anyway, I hope the authors will address my concerns.”

Answer 3:

We understand that the reviewer is referring specifically to the caspase 1 western blot images, which are the ones that initially were initially deemed inadequate by the reviewer. Obviously, the reviewer is right that increasing the exposure time causes the bands to become more saturated, but after reviewing all the images we had, we don't have one exactly as the reviewer says. In our laboratory, we tend not to expose Western blots for long periods of time because we prefer clearer band images. For this reason, we repeated the western blot and used another antibody (Caspase-1 14F468, sc-56036 Santa Cruz Biotechnology) in order to better identify the band with a molecular size of 45 kDa and obtain a better, more publishable image (mature caspase-1). We believe that the result has been optimal, as the reviewer can see in the new image in Figure 5.

The problem with using another antibody is that it only recognized the mature form, and we were unable to see the band between 10-20 kDa corresponding to the activated form. Therefore, we have left the first band that was in the figure after the review. However, if the reviewer still thinks it is not appropriate, we can remove it if necessary in the next review.

With regard to the intensity of the VDAC bands, as we have mentioned, we do not like to overexpose the bands precisely in order to see the correct differences.

We apply the same principle to the VDAC band, where some of the differences that may be found may also be due to pipetting errors, a consideration that is applicable to any housekeeping, where small differences between bands are always found.
